# KAT6A is essential for developmental control gene expression in neural stem and progenitor cells

Anne K. Voss[1,2]*, Samantha Eccles[1,2], Johannes Wichmann[1,2], Waruni Abeysekera[1,2], Maria I. Bergamasco[1,2], Alexandra L. Garnham[1,2], Nishika Ranathunga[1,2], Yuqing Yang[1,2], Rory Bowden[1,2], Gordon K. Smyth[1,3], Tim Thomas[1,2]*

1 The Walter and Eliza Hall Institute of Medical Research, Melbourne, Victoria, Australia, 2 Department of Medical Biology, University of Melbourne, Melbourne, Victoria, Australia, 3 School of Mathematics and Statistics, University of Melbourne, Parkville, Victoria, Australia

* avoss@wehi.edu.au (AKV); tthomas@wehi.edu.au (TT)

## Abstract

Heterozygous variants in the *KAT6A* gene encoding the histone lysine acetyltransferase KAT6A (MOZ, MYST3) cause Arboleda-Tham syndrome, a cognitive impairment syndrome. Histone acetylation is generally associated with active gene transcription. Genetic deletion of both alleles of the *Kat6a* gene in mice causes developmental defects including anterior homeotic transformation, cleft palate, interrupted aortic arch and cardiac septal defects. Loss of KAT6A impairs expression of HOX, DLX and TBX genes, which are essential for body segment identity specification, palate, heart and aortic arch development. However, the effects of loss of KAT6A on chromatin modifications and gene expression in neural cells, which are relevant to normal brain development and function, is still poorly understood. In this study, we used an automated high-throughput chromatin profiling method and RNA sequencing in mouse neural system and progenitor cells to assess the effects of loss of one or two alleles of *Kat6a* on gene expression, histone acetylation and methylation. We also assessed occupancy by a trithorax group protein and RNA polymerase II. Our data suggests two modes of action for KAT6A: (1) acetylation of histone H3 on lysine 23 at promoters and enhancers and (2) recruitment of the trithorax group protein MLL1 (KMT2A) to promote the expression of developmental genes, including SOX and homeodomain genes. Together, these two functions appear to be required for normal gene expression in neural progenitors and essential for proliferation and neuronal differentiation.

## Author summary

During embryonic development, specific families of transcription factors pattern the early embryo to lay down and define the body and organ structure. However,

**Data availability statement:** RNA sequencing data have been submitted to the NCBI Gene Expression Omnibus (GEO accession number: GSE314079). CUT&Tag sequencing data have been submitted to GEO (GEO accession number: GSE311724). Gene expression levels and fold-changes of all genes detected are available in S1 to S4 Tables. CUT&Tag results for differential occupancy of the genome for all genes detected are available in S5 and S6 Tables.

**Funding:** This work was supported by the Lorenzo and Pamela Galli Medical Research Trust to AKV; the Valda Klaric Foundation to TT; the Australian National Health and Medical Research Council through project grant 1160517 to TT, Ideas Grant 2010711 to TT, Research Fellowships 1081421 to AKV and 1154970 to GKS, and Investigator Grants 1176789 to AKV and 2025645 to GKS; through the Independent Research Institutes Infrastructure Support Scheme; and by the Victorian Government through an Operational Infrastructure Support Grant. The funders had no role in study design, data collection and analysis, decision to publish, or preparation of the manuscript.

**Competing interests:** I have read the journal's policy and the authors of this manuscript have the following competing interests: AKV and TT are inventors on patent WO2016198507A1. AKV and TT have received research funding from the Cancer Therapeutics CRC (CTX). AKV and TT have served on a clinical advisory board for Pfizer. The other authors have declared that no competing interests exist.

the mechanisms governing the onset of the expression of these developmental transcription factors in less well understood. KAT6A is thought to promote gene expression by acetylation histone proteins. Here we determine the effects of KAT6A on histone acetylation and gene expression in mouse neural stem and progenitor cells. Our data are relevant for the understanding of pathogenic genetic variants in one allele of the human *KAT6A* gene, which cause the Arboleda-Tham cognitive impairment syndrome.

## Introduction

The lysine (K) acetyltransferase KAT6A is one of nine nuclear histone lysine acetyltransferases encoded by the mammalian genome that have structurally defined acetyltransferase domains [1,2]. Heterozygous pathogenic variants in the *KAT6A* gene cause Arboleda-Tham syndrome, characterised by global developmental delay and cognitive dysfunction (93% penetrance), speech delay (73%), visual impairment (77%), gastrointestinal dysfunction (69%), sleep disruption (65%), congenital heart defects (51%; including septal defects), frequent infections (47%) and autism-like behaviour (32%), among other less common anomalies [3–6]. Most pathogenic variants (90% to 92%) in *KAT6A* are truncating mutations [5,6]. Variants truncating the KAT6A protein in more carboxy-terminal regions appear to be associated with more severe anomalies [5]. Consistent with heterozygous pathogenic variants in the *KAT6A* gene causing the Arboleda-Tham cognitive impairment syndrome, *Kat6a*$^{+/-}$ mice display learning and memory defects [7,8], as well as hyperactivity and autism-like sociability defects [7].

Histone acetylation is generally more abundant at transcriptionally active gene loci [9] and thought to promote gene expression [10,11]. KAT6A is a member of the MYST family of histone acetyltransferases, which comprises five of the nine nuclear histone acetyltransferases [1,2]. KAT6A occurs as the catalytic component in a large multi-protein complex [12] that, besides KAT6A or its paralogue KAT6B, contains the adaptor proteins BRPF1, 2 or 3 and ING4 or 5, as well as MEAF6 [12]. BRPF1, 2 and 3 bind H2AK5ac, H4K12ac and H3K14ac via bromodomains and unmodified histone H3 tails and nucleosomal DNA via plant homeodomain (PHD) fingers [13,14]. ING4 and 5 bind histone H3 methylated on lysine 4 via their ING domains [15–17]. Comparison of the loss of function phenotypes between the protein complex members [18–28] and protein-protein interactions studies involving overexpression in cultured cells [29,30] suggest that the most prominent 'KAT6' protein complex composition during development is KAT6A (or alternatively KAT6B) with BRPF1, ING5 and MEAF6.

Homozygous null mutation of the *Kat6a* gene in mice causes an extensive anterior homeotic transformation of 19 skeletal and neural body segments and a posterior shift in HOX gene expression as well as lowered HOX gene expression [25]. Body segment identity is conferred by the sequential, collinear activation and nested expression of HOX genes in a specific anterior to posterior time course and with

specific anterior HOX gene expression boundaries [reviewed in [31–33]]. Activation of HOX gene expression by treating pregnant females with retinoic acid rescues the *Kat6a* knockout body segment identity defect [25]. The time course and expression boundaries of HOX genes are controlled by the antagonistic action of polycomb repressor complexes and trithorax group proteins [reviewed by [34–36]]. For example, mutation of the polycomb repressor complex 1 protein, BMI1 (also known as PCGF4 or RNF51) leads to a posterior homeotic transformation [37]. Congruent with a role for KAT6A in activating HOX genes, combined loss of KAT6A and BMI1, rescues both the *Kat6a* and the *Bmi1* knockout body segment identity defects and restores the anterior HOX gene expression boundaries to normal [23]. Furthermore, loss of KAT6A causes cleft palate and reduced expression of DLX genes [38,39] and aortic arch and cardiac septal defects along with reduced expression of TBX genes [39,40]. Transgenic overexpression of *Tbx1* rescues the *Kat6a* knockout aortic arch defects [39]. Moreover, deletion of *Kat6a* in the germline or in adult haematopoietic cells leads to the complete absence of haematopoietic stem cells [19,24,41]. These observations demonstrate the critical roles that KAT6A plays during development of different organ systems.

The deletion of the *Kat6a* gene results in a reduction of histone H3 lysine 9 acetylation (H3K9ac) at HOX, DLX and TBX genes, among other genes [25,38,39], but no global genome-wide changes in H3K9 or H3K14 acetylation levels [25]. In addition, a reduction in the occupancy of HOX genes by the trithorax group protein mixed lineage leukaemia I [(MLL1), also known as histone-lysine N-methyltransferase 2A (KTM2A)] was observed in the absence of KAT6A [25]. Deletion of one or two alleles of *Kat6a* results in a global reduction in H3K23ac in mouse embryos ex vivo [42], in adult mouse brain cells [8], in adult mouse brain and in adult mouse peripheral white blood cells [7]. Knockdown of *KAT6A* causes a reduction in H3K23ac in cultured human cells [43,44], while lowerH3K14ac has also been reported [12]. Introduction of Arboleda-Tham syndrome mutations in *KAT6A* gene exons 2–15 in cultured human cells cause a reduction in H3K23ac [7]. Together, these studies suggest that KAT6A may promote gene expression by acetylating histone H3 at lysines 9, 14 and 23. Alternatively, some of the changes in histone acetylation levels may be secondary effects.

Pathogenic variants in only one allele of *KAT6A* have profound effects on cognitive development and function. This raises the question of how heterozygous and homozygous loss of *KAT6A* affect gene expression and chromatin modifications, specifically in a neural cell type derived from the developing brain that is functionally affected by loss of KAT6A. To address this question, we established that murine embryonic forebrain neural stem and progenitor cells form fewer neurons after loss of one and two alleles of *Kat6a* and used these cells to conduct automated chromatin profiling. We determined within the same cell preparation acetylation levels at histone H3 lysines 9, 14 and 23, as well as RNA polymerase II (POLR2A) and MLL1 occupancy and methylation levels at histone H3 lysine 4 (H3K4), the MLL1 target. We compared these chromatin profiling results to the effects of loss of KAT6A on gene expression.

## Results

To examine the effects of homozygous and heterozygous mutation of *Kat6a* on gene expression and chromatin modifications in cells derived from the developing brain, we isolated neural stem and progenitor cells (NSPCs) from *Kat6a*+/+ wildtype, *Kat6a*+/− heterozygous and *Kat6a*−/− homozygous mutant E12.5 mouse dorsal telencephalon, the precursor of the cerebral cortex. E12.5 was chosen, because *Kat6a*−/− mouse embryos typically die at E13.5 on a C57BL/6 genetic inbred background.

### Lack of *Kat6a* affected global levels of H3K23ac and neuronal differentiation

We determined the effects of loss of KAT6A on histone acetylation at histone H3 lysine 9, 14 and 23 by comparing acetylation in *Kat6a*+/+ and *Kat6a*−/− NSPCs. These are the previously published targets in a variety of human cell types and mouse embryos [12,25,39,43,44]. Global histone H3 lysine 23 acetylation (H3K23ac) was moderately, but consistently reduced in *Kat6a*−/− vs. *Kat6a*+/+ NSPCs (Fig 1A and 1B), whereas genome-wide acetylation levels at H3K9 and H3K14 were unchanged (S1A - S1D Fig). Examination of NSPCs cultured under proliferation conditions in vitro revealed

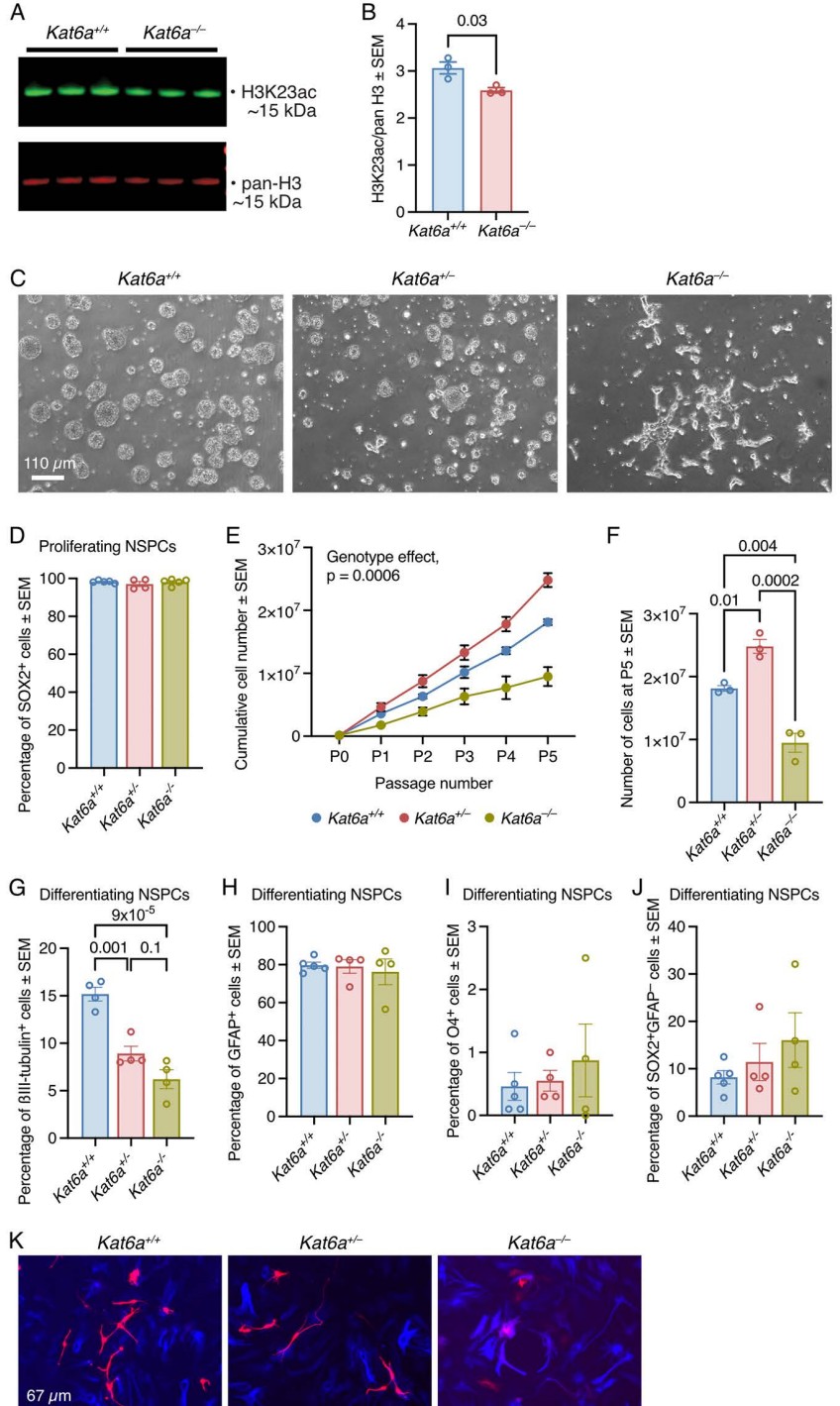

**Fig 1. Loss of KAT6A causes a modest reduction in H3K23ac, as well as a reduction in proliferation and neuronal differentiation in neural stem and progenitor cells (NSPCs).** (A) H3K23 acetylation levels and pan histone H3 levels assessed by Western blotting (A) and densitometry (B) in *Kat6a*⁺/⁺ wildtype and *Kat6a*⁻/⁻ homozygous mutant NSPCs. Each lane was loaded with 0.5 μg of acid extracted protein from NSPCs isolated from an individual mouse embryo. H3K23ac levels were normalised to pan-H3 levels. (C) Representative phase contrast images of proliferating *Kat6a*⁺/⁺ wildtype, *Kat6a*⁺/⁻ heterozygous mutant and *Kat6a*⁻/⁻ homozygous mutant NSPCs. (D) Percentage of cells expressing the neural stem cell (NSC) marker protein SOX2 assessed by intranuclear immunofluorescence staining and flow cytometry in proliferating *Kat6a*⁺/⁺, *Kat6a*⁺/⁻ and *Kat6a*⁻/⁻ NSPCs. (E,F) Cumulative

cell count of *Kat6a^+/+*, *Kat6a^+/−* and *Kat6a^−/−* NSPCs grown over 5 passages (P1 to P5) in culture (E) and cumulative cell number of cells at passage 5 (F). (G-J) Percentage of cells expressing the neuronal marker protein ßIII-tubulin (G), the astrocyte marker glial acidic protein (GFAP; H), the oligodendrocyte epitope O4 (I) and the NSC marker protein SOX2 but not GFAP (J) assessed by immunofluorescence staining and flow cytometry of *Kat6a^+/+*, *Kat6a^+/−* and *Kat6a^−/−* NSPCs after three days of culture in differentiation conditions. (K) Representative epifluorescence images of *Kat6a^+/+*, *Kat6a^+/−* and *Kat6a^−/−* NSPCs after five days of differentiation stained for ßIII-tubulin (red) and GFAP (blue). NSPC isolates from N = 3 (A-C,E,F) and 4 (D,G-K) E12.5 embryos per genotype. Each circle in (B,D,F-J) represents NSPCs isolated from an individual mouse embryo. Data are presented as mean ± SEM and were analysed by unpaired, two-tailed Student's t-test (B), one-way ANOVA with Tukey's multiple comparison test (D,F-J) and two-way ANOVA with Tukey's multiple comparison test (E). Scale bars in (C) equals 110 µm and in (K) 67 µm.

the expected appearance of *Kat6a^+/+* wildtype NSPCs as spherical, floating colonies (neurospheres) ([Fig 1C]). In contrast, *Kat6a^+/−* NSPCs formed smaller colonies that tended to attach and *Kat6a^−/−* NSPCs formed few floating colonies and instead attached to the uncoated tissue culture plastic ([Fig 1C]). However, NSPCs of all three genotypes robustly expressed the neural stem cell marker protein SOX2 ([Fig 1D]). *Kat6a^−/−* NSPCs proliferated more slowly than the other two genotypes ([Fig 1E] and [1F]).

Congruent with our results, a neural stem cell proliferation defects were previously observed in embryos lacking the histone acetyltransferase function of KAT6A [45] and neural stem and precursor cell deficiencies have been observed in mice after forebrain-specific deletion of the gene encoding the KAT6A protein complex protein BRPF1 [46].

Cultured for 3 days under differentiating conditions, *Kat6a^+/−* and *Kat6a^−/−* NSPCs formed fewer cells expressing the neuronal marker ßIII-tubulin compared to *Kat6a^+/+* NSPCs (p = 0.001 and 9x10^-5, respectively); proportions of GFAP and O4 positive cells, as well as SOX2 positive and GFAP negative cells were not significantly different between genotypes ([Fig 1G]–[1K]). These data suggest that the presence of KAT6A is required for normal neuronal differentiation.

## Normal expression of developmental control genes requires KAT6A

Using proliferating and differentiating *Kat6a^+/+*, *Kat6a^+/−* and *Kat6a^−/−* NSPCs for RNA sequencing, we observed thousands of changes in mRNA levels between *Kat6a^−/−* and *Kat6a^+/+* NSPCs (FDR < 0.05; [Fig 2A]–[2D]). In proliferating NSPCs, 3718 genes were differentially expressed in *Kat6a^−/−* vs. *Kat6a^+/+* NSPCs (2017 genes up and 1701 genes down; [Fig 2A] and [S1 Table]) and in differentiating NSPCs, 5691 genes were differentially expressed in *Kat6a^−/−* vs. *Kat6a^+/+* NSPCs (2810 genes up and 2881 genes down; [Fig 2B] and [S2 Table]). Unsupervised hierarchical clustering revealed that *Kat6a^−/−* and *Kat6a^+/+* NSPCs tended to cluster within genotype and that gene expression in *Kat6a^+/−* NSPCs was positioned intermediate between *Kat6a^−/−* and *Kat6a^+/+* NSPCs ([Fig 2C] and [2D]). Multidimensional scaling also showed that proliferating NSPCs, and to a lesser extent differentiating, *Kat6a^+/−* NSPCs, were positioned intermediate between *Kat6a^−/−* and *Kat6a^+/+* NSPCs ([S2A] and [S2B Fig]).The top 20 genes downregulated in proliferating *Kat6a^−/−* vs. *Kat6a^+/+* NSPCs contained the neural fate determining gene *Sox1* on position 3 (FDR = 3x10^-5; [Fig 2E]). Among the top genes downregulated in differentiating *Kat6a^−/−* vs. *Kat6a^+/+* NSPCs, 14 of 20 genes were required for nervous system development (*Esco2, Knl1, Eomes*, *Kif14*, *Insm1*, *Dcc*, *Cadps*, *Hes5*, *Ankle2*, *Elavl4*, *Pkmyt*, *Dcx*, *Dll3*, and *Myc*; FDR = 2x10^-5 to 0.0009; [Fig 2F]). Gene families that were overall downregulated in the absence of *Kat6a* included the SOX gene family in differentiating *Kat6a^−/−* vs. *Kat6a^+/+* NSPCs (FDR = 6x10^-5 to 0.02; [Fig 2G]). *Sox1* was also downregulated in proliferating *Kat6a^−/−* vs. *Kat6a^+/+* NSPCs (FDR = 3x10^-5; [S2C Fig]). In addition, the DLX gene family (FDR = 0.0009 to 0.02; [S2D Fig]) and MEIS genes were downregulated in differentiating *Kat6a^−/−* vs. *Kat6a^+/+* NSPCs (FDR = 4x10^-5 and 0.001; [S2E Fig]). Notable was the number of homeodomain protein encoding genes that were downregulated in differentiating *Kat6a^−/−* vs. *Kat6a^+/+* NSPCs (FDR = 3x10^-5 to 0.02; [Fig 2H]). In contrast, genes upregulated in proliferating or differentiating *Kat6a^−/−* vs. *Kat6a^+/+* NSPCs included protocadherin genes ([S2F Fig]) and genes important for DNA repair and nucleic acid metabolic processes ([S2G] and [S2H Fig]). Only minor effects were observed on genes encoding other KAT6A protein complex members and other histone lysine acetyltransferases ([S2I] – [S2M Fig]). Congruent with the reduced cell proliferation observed in *Kat6a^−/−* vs. *Kat6a^+/+* NSPCs, gene ontology term analysis revealed that genes important for cell division and cycle phase transition were downregulated in

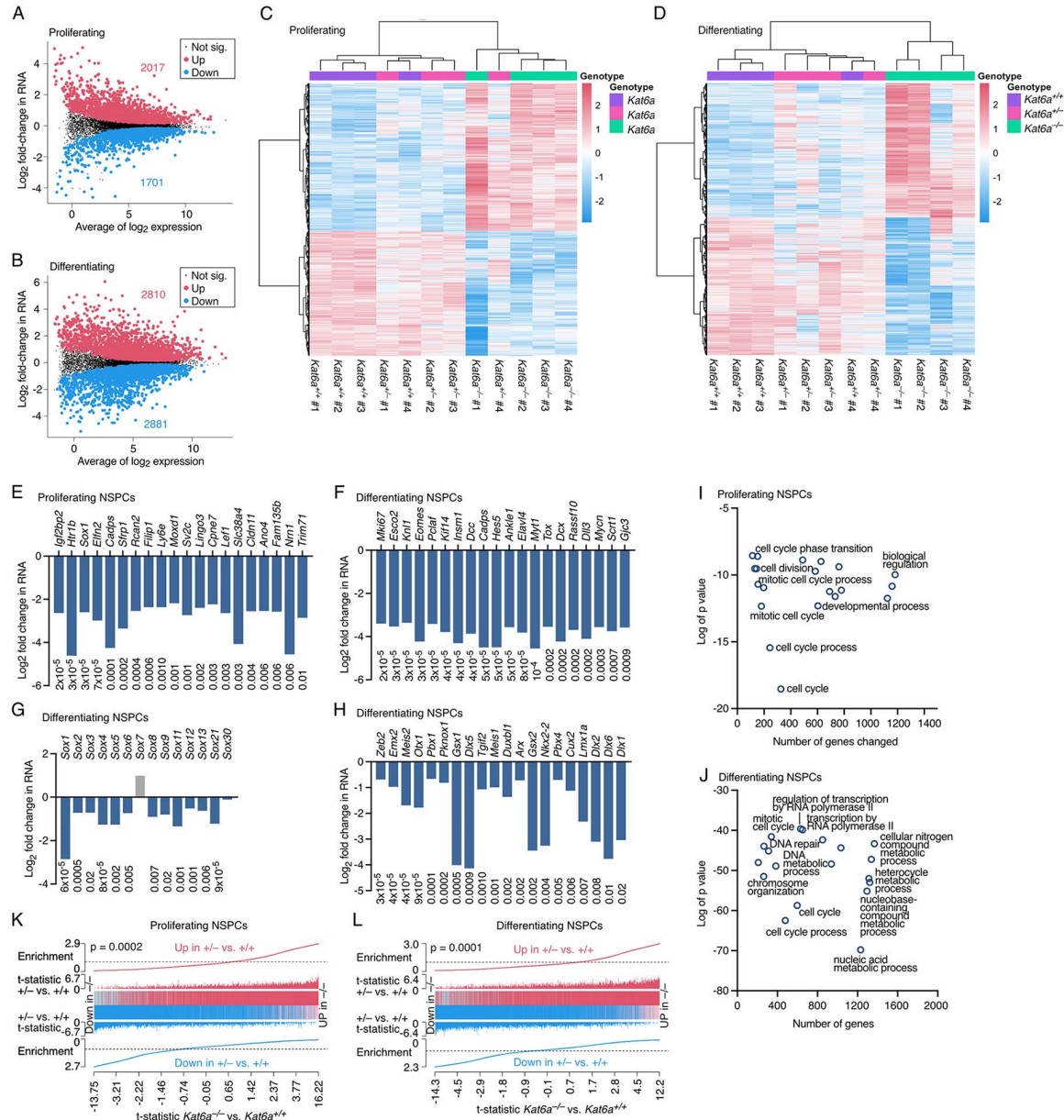

**Fig 2. Loss of KAT6A causes a reduction in the expression of developmental control genes.** (A-L) RNA sequencing data of NSPCs isolated from N = 4 $Kat6a^{+/+}$, 4 $Kat6a^{+/-}$ and 4 $Kat6a^{-/-}$ E12.5 embryos. Data were analysed as described in the methods section under RNA sequencing data analysis. Differences in gene expression with a false discovery rate (FDR) < 0.05 were considered significant. (A,B) M ($\log_2$ ratio) and A (mean average) plot of genes differentially expressed in $Kat6a^{-/-}$ vs. $Kat6a^{+/+}$ NSPCs grown under proliferating (A) and differentiating conditions (B) for 3 days. The number of upregulated (red) and downregulated genes (blue) are indicated. Genes not significantly changed are indicated in black. (C,D) Heatmaps showing hierarchical clustering of samples and genes differentially expressed in the contrast of proliferating (C) and differentiating (D) $Kat6a^{-/-}$ vs. $Kat6a^{+/+}$ NSPCs but also displaying results for $Kat6a^{+/-}$ NSPCs. (E,F) $\log_2$ fold-change in RNA levels of the top 20 genes (by fold-change amplitude, average expression > 1 CPM) downregulated in proliferating (E) and differentiating (F) $Kat6a^{-/-}$ vs. $Kat6a^{+/+}$ NSPCs. FDRs shown below the bars. (G,H) $\log_2$ fold-change in RNA levels of SOX genes (G) and homeodomain protein genes (H) in differentiating $Kat6a^{-/-}$ vs. $Kat6a^{+/+}$ NSPCs. FDRs shown below the bars. (I,J) Top 20 gene ontology terms (biological process) associated with genes downregulated in proliferating (I) and differentiating (J) $Kat6a^{-/-}$ vs. $Kat6a^{+/+}$ NSPCs. (K,L) Barcode plots showing significant positive correlations between gene expression changes in proliferating (K; p = 0.0002) and differentiating (L; p = 0.0001) $Kat6a^{-/-}$ vs. $Kat6a^{+/+}$ and $Kat6a^{+/-}$ vs. $Kat6a^{+/+}$ E12.5 NSPCs. The horizontal axis shows t statistics for all genes in the $Kat6a^{-/-}$ vs. $Kat6a^{+/+}$ dataset. The long vertical lines represent genes and the shorter vertical lines the t-statistics of genes in the $Kat6a^{+/-}$ vs. $Kat6a^{+/+}$ dataset. Worms show the relative enrichment of the genes up (red) and downregulated (blue) in the $Kat6a^{+/-}$ vs. $Kat6a^{+/+}$ dataset.

proliferating *Kat6a*⁻/⁻ vs. *Kat6a*⁺/⁺ NSPCs (Fig 2I and S1 Table). Cell cycle genes were also downregulated in differentiating *Kat6a*⁻/⁻ vs. *Kat6a*⁺/⁺ NSPCs (Fig 2J and S2 Table) and genes required for RNA polymerase II mediated transcription, DNA repair and nucleobase metabolic processes were downregulated (Fig 2J and S2 Table). More specific to the brain, genes downregulated in differentiating *Kat6a*⁻/⁻ vs. *Kat6a*⁺/⁺ NSPCs were involved in brain, and explicitly forebrain, development and included genes encoding DNA-binding transcription factors (all $p < 10^{-6}$; S2 Table). Even proliferating *Kat6a*⁻/⁻ vs. *Kat6a*⁺/⁺ NSPCs already displayed a decrease in brain and forebrain development genes ($p = 0.0004$ and $0.003$, respectively), as well as in neural precursor cell proliferation genes ($8 \times 10^{-5}$; S1 Table).

The analysis of proliferating and differentiating *Kat6a*⁺/⁻ vs. *Kat6a*⁺/⁺ NSPCs did not return any differentially expressed individual genes that reached $FDR < 0.05$ significance (S3 and S4 Tables). However, gene set analysis showed that gene expression changes in *Kat6a*⁺/⁻ vs. *Kat6a*⁺/⁺ NSPCs correlated positively and significantly with genes expression changes in *Kat6a*⁻/⁻ vs. *Kat6a*⁺/⁺ NSPCs, both, for the proliferating and the differentiating condition ($p = 0.0002$ and $0.0001$, respectively; Fig 2K and 2L), suggesting that a multitude of modest changes in gene expression might contribute to the dysfunctions observed in individuals with heterozygous pathogenic variants in the *KAT6A* gene and in *Kat6a*⁺/⁻ mice.

## MLL1 gene occupancy and H3K4 trimethylation are reduced in the absence of KAT6A

We performed automated CUT&Tag chromatin profiling [47] on NSPCs isolated from the dorsal telencephalon (the cerebral cortex precursor) of *Kat6a*⁺/⁺, *Kat6a*⁺/⁻ and *Kat6a*⁻/⁻ E12.5 embryos, to detect multiple chromatin marks in the same biological cell sample. These included the acetylation targets proposed for KAT6A, namely H3K9ac, H3K14ac and H3K23ac, as well as H3K4me3, MLL1 and RNA polymerase II occupancy. We had previously observed MLL1 occupation of HOX genes to be dependent on the presence of KAT6A in whole E10.5 embryos [25]. Thus, we included MLL1. Within its four to five subunit protein complex, MLL1 has H3K4me2/3 methyltransferase activity [48–51]. We therefore assessed H3K4me3, which is the target of MLL1 that is most closely associated with mRNA levels [52]. In addition, we included RNA polymerase II subunit A phosphorylated on Ser2 (POLR2A; also known as Phospho-Rpb1 CTD). Phosphorylation of POLR2A at Ser2 is required for elongation and allowed us to determine if the absence of KAT6A affected POLR2A progression through a gene locus.

In differentiating NSPCs, the loss of one or two alleles of *Kat6a* caused a significant dose-depended reduction in the read count aggregated over all gene bodies for H3K9ac, H3K23ac, MLL1 and H3K4me3 (Fig 3A). Similar reductions were seen in promoters (Figs 3B to 3E and S3A), enhancers and, specifically, active enhancers in differentiating NSPCs (S3B and S3C Fig). H3K14ac was not noticeably reduced by loss of *Kat6a* alleles in differentiating NSPC (Figs 3A and S3A – S3D). Curiously, POLR2A appeared to be slightly increased with the loss of *Kat6a* alleles (Figs 3A, S3A – S3C, and S3E), which might suggest that POLR2A progressed more slowly through gene loci in differentiating *Kat6a*⁻/⁻ NSPCs compared to *Kat6a*⁺/⁺ cells. Genotype effects were less pronounced in proliferating NSPCs (S3F – S3I Fig). H3K9ac was reduced at promoters, enhancers, active enhancers and gene bodies in proliferating *Kat6a*⁻/⁻ vs. *Kat6a*⁺/⁺ NSPCs and not in *Kat6a*⁺/⁻ NSPCs (S3F to S3I Fig). H3K14ac and H3K23ac were reduced in a gene dose dependent manner in gene bodies (Fig 3I). H3K4me3 was reduced in a *Kat6a* gene dose dependent manner at promoters, enhancers, active enhancers and gene bodies in proliferating NSPCs (S3F to S3I Fig). However, MLL1 occupancy was not reduced in the absence of KAT6A in proliferating NSPCs (S3F to S3I Fig), suggesting that another SET domain protein, that may be affected by loss of KAT6A, might contribute to H3K4me3 in proliferating NSPCs. POLR2A occupancy was largely unaffected by the absence of KAT6A in proliferating NSPCs (S3F to S3I Fig). Examination of read depth plots from 1 kb upstream of the TSS to 1 kb downstream of the transcription end site (TES) showed that differences between genotypes were observed across the promoters and bodies of genes (S3J to S3M Fig).

In addition to examining the read count aggregated over all genes (Figs 3A–3E and S3A -S3M), we assessed the read count aggregates separately over genes that were upregulated and genes that were downregulated in proliferating and differentiating *Kat6a*⁻/⁻ vs. *Kat6a*⁺/⁺ NSPCs (S4 Fig). The results over upregulated and downregulated genes (S4A -S4P Fig)

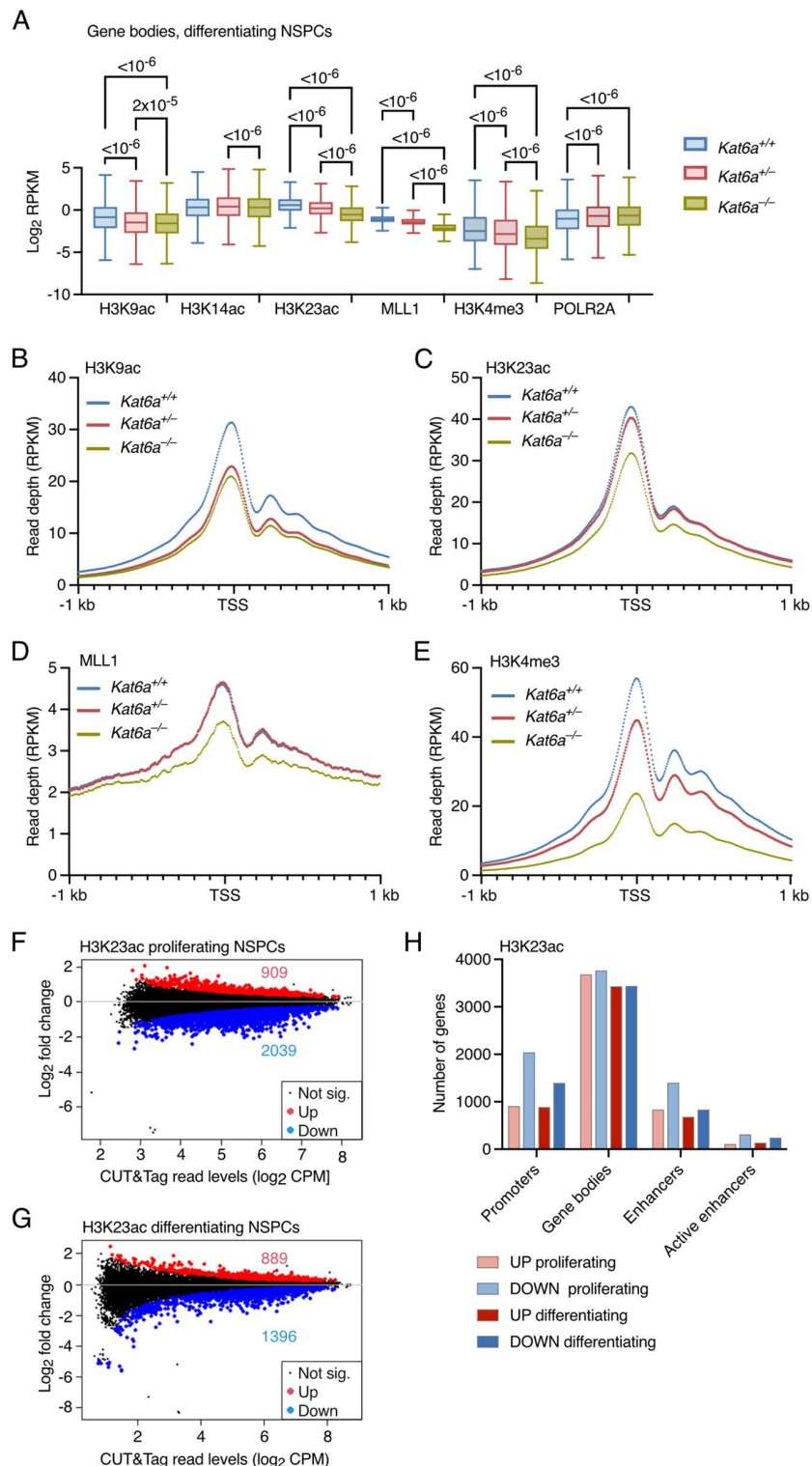

**Fig 3. Loss of KAT6A causes reductions in H3K9ac, H3K23ac and H3K4me3 levels as well as a reduction in MLL1 occupancy.** (A-H) CUT&Tag results of NSPCs isolated from N = 3 *Kat6a*⁺/⁺, 4 *Kat6a*⁺/⁻ and 3 *Kat6a*⁻/⁻ E12.5 embryos. Data were analysed as described in the methods section under Automated CUT&Tag sequencing data analysis. Differences in occupancy with a false discovery rate (FDR) < 0.05 were considered significant. Data

in (A) were analysed by Kruskal-Wallis test followed by Dunn's correction for multiple testing. (A) Log$_2$ of CUT&Tag read count per kilobase normalised to library size (RPKM) accrued over gene bodies for H3K9ac, H3K14ac, H3K23ac, MLL1, H3K4me3 and POLR2A in differentiating *Kat6a*$^{+/+}$, *Kat6a*$^{+/-}$ and *Kat6a*$^{-/-}$ NSPCs. (B-E) Read depth aggregates over all protein coding genes for H3K9ac (B), H3K23ac (C), MLL1 (D) and H3K4me3 (E) over the interval from -1 kb to +1 kb of the transcription start site (TSS) in differentiating *Kat6a*$^{+/+}$, *Kat6a*$^{+/-}$ and *Kat6a*$^{-/-}$ NSPCs. (F,G) MA plot of genes differentially decorated with H3K23ac in promoters of proliferating (F) and differentiating (G) *Kat6a*$^{-/-}$ vs. *Kat6a*$^{+/+}$ NSPCs. (H) Number of genomic regions displaying differences in levels of H3K23ac in proliferating and differentiating *Kat6a*$^{-/-}$ vs. *Kat6a*$^{+/+}$ NSPCs.

were similar to the read count aggregated over all genes (Figs 3A–3E, S3A, S3B, S3D – S3F, and S3I). Only a modestly stronger genotype effect of the reduction in histone acetylation, MLL1 occupancy, H3K4me3 and POLR2A occupancy was observed in the downregulated as compared to the upregulated genes (S4Q – S4T Fig).

While the read count aggregated over all genes provides an insight into major changes in the chromatin landscape, individual genes may be affected in a more specific manner.

### H3K23ac is reduced in the absence of KAT6A

The largest number of significant changes at the level of individual genes were observed in H3K23ac levels, changed in the body of the genes of 7448 individual genes in proliferating *Kat6a*$^{-/-}$ vs. *Kat6a*$^{+/+}$ NSPCs (3683 increased and 3765 decreased) and at 6873 individual genes in differentiating *Kat6a*$^{-/-}$ vs. *Kat6a*$^{+/+}$ NSPCs (3434 increased and 3439 decreased; S5, S6, and S7 Tables). However, a skewing towards a reduction in the acetylation mark is what one would expect for results including direct targets of KAT6A. A predominant decrease in proliferating *Kat6a*$^{-/-}$ vs. *Kat6a*$^{+/+}$ NSPCs was observed for H3K23ac levels at active enhancers (312 genes decreased in H3K23ac and 110 genes increased; S5 and S7 Tables), promoters (2039 down, 909 up; Fig 3F and S5 and S7 Tables) and all enhancers (1402 down, 835 up; S5 and S7 Tables). In addition, H3K9ac, H3K14ac and H3K4me3 were skewed towards a reduction particularly at active enhancers, albeit in a smaller number of genes (S5 and S7 Tables).

Like proliferating NSPCs, differentiating *Kat6a*$^{-/-}$ vs. *Kat6a*$^{+/+}$ NSPCs also showed the greatest skewing towards a reduction in H3K23ac in promoters (1396 down, 889 up; Fig 3G and S6 and S7 Tables) and active enhancers (241 down, 137 up; S6 and S7 Tables). While significant changes were observed for other chromatin marks in differentiating *Kat6a*$^{-/-}$ vs. *Kat6a*$^{+/+}$ NSPCs, these were not obviously skewed with respect to the number of genes showing an increase vs. a decrease in individual chromatin marks (S6 and S7 Tables).

Overall, we observed a skewing towards a decrease in H3K23ac in the absence of KAT6A. This was observed across a large number of individual genes in both proliferating and differentiating NPSCs, in particular at promoters and enhancers (Fig 3H). This, however, does not preclude that other, less numerous changes are functionally relevant.

### Changes in MLL1 occupancy and H3K4 trimethylation correlates strongly with changes in gene expression in the absence of KAT6A

To determine relationships between the effects on gene transcription and chromatin marks, we assessed the correlation between changes in chromatin marks and RNA levels at individual genes in *Kat6a*$^{-/-}$ vs. *Kat6a*$^{+/+}$ NSPCs.

We observed the tightest positive correlation between the change in MLL1 occupancy at promoters and change in RNA levels per gene in proliferating *Kat6a*$^{-/-}$ vs. *Kat6a*$^{+/+}$ NSPCs Fig 4A). MLL1 occupancy ($R^2 = 0.94$) and H3K4me3 ($R^2 = 0.75$) were decreased at the promoters of neural stem cell and early brain development genes (Fig 4A and 4B) and in the gene bodies ($R^2 = 0.62$ and 0.81, respectively), including at the *Sox1*, *Otx1, Emx2* and *Dmrta2* genes in the absence of KAT6A (Fig 4C–4E). Congruently, mRNA output of these genes was downregulated significantly in *Kat6a*$^{-/-}$ vs. *Kat6a*$^{+/+}$ NSPCs. In *Kat6a*$^{+/-}$ vs. *Kat6a*$^{+/+}$ NSPCs, these genes only showed a tendency for downregulation (Fig 4F and S1 to S4 Tables). The other chromatin marks that showed a positive correlation between changes in the chromatin mark and changes in RNA levels in proliferating *Kat6a*$^{-/-}$ vs. *Kat6a*$^{+/+}$ NSPCs at specific gene promoters and bodies

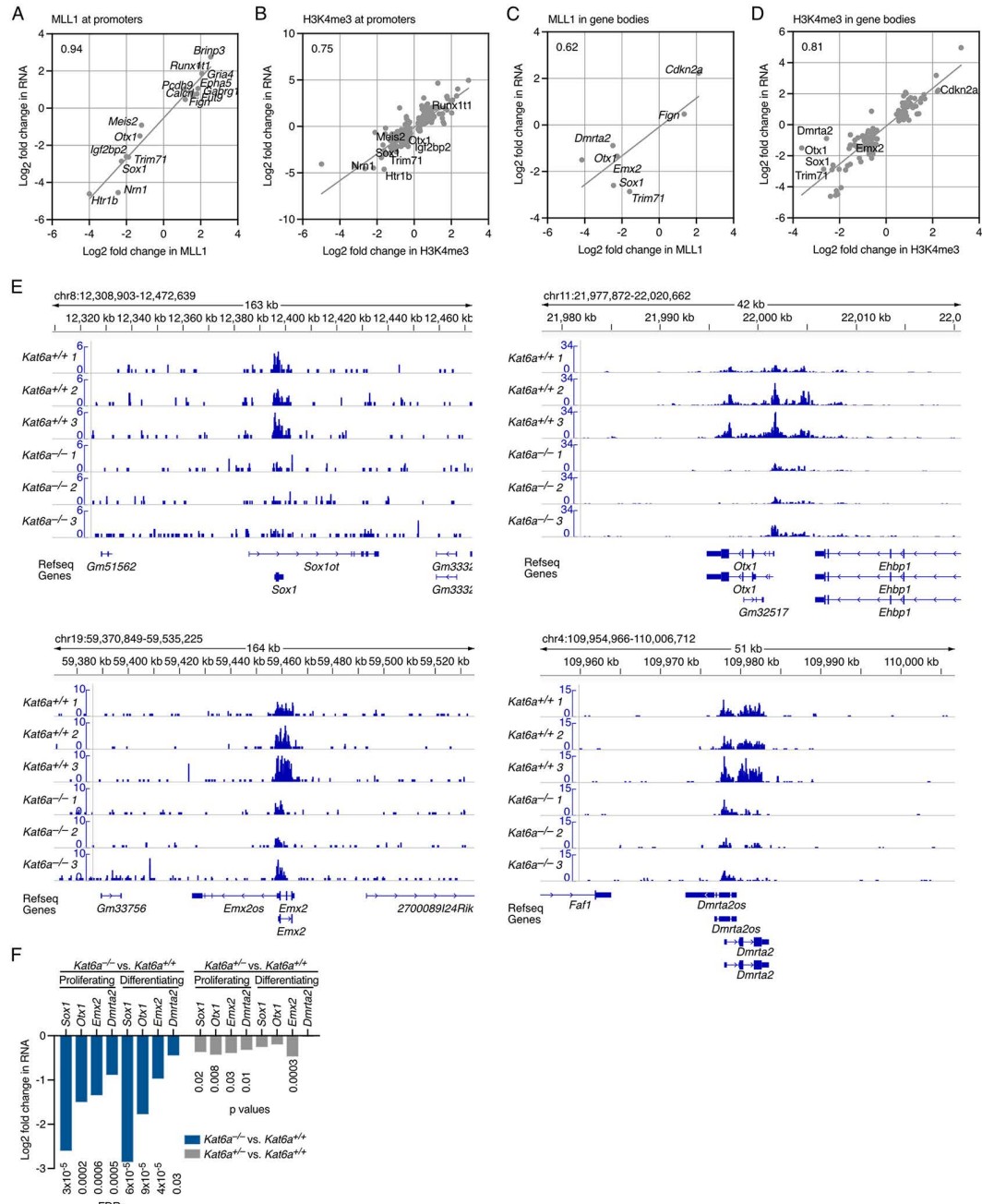

**Fig 4. Loss of KAT6A results in a reduction in MLL1 and H3K4me3 associated with a reduction in mRNA levels.** (A-D) Comparison of results from RNA sequencing and CUT&Tag results of NSPCs isolated from N = 3–4 *Kat6a*[+/+] and 3–4 *Kat6a*[−/−] E12.5 embryos. Data were analysed as described in the methods section under RNA sequencing data analysis and Automated CUT&Tag sequencing data analysis. Differences in gene expression or occupancy with a false discovery rate (FDR) < 0.05 were considered significant. (A-D) Correlation between $\log_2$ fold-change in RNA levels and $\log_2$ fold-change in MLL1 occupancy (A,C) or H3K4me3 levels (B,D) of proliferating *Kat6a*[−/−] vs. *Kat6a*[+/+] NSPCs at promoters (A,B) and bodies (C,D) of protein coding genes differentially expressed at FDR < 0.05 and differentially occupied at FDR < 0.05. (E) Read depth plots of CUT&Tag results for MLL1 occupancy at the *Sox1*, *Otx1*, *Emx2* and *Dmrta2* locus. (F) $\log_2$ fold-change in RNA levels of *Sox1*, *Otx1*, *Emx2* and *Dmrta2* in proliferating and differentiating *Kat6a*[−/−] vs. *Kat6a*[+/+] and *Kat6a*[+/−] vs. *Kat6a*[+/+] NSPCs.

PLOS Genetics

were H3K9ac ($R^2 = 0.69$ and 0.69, respectively), H3K14ac ($R^2 = 0.82$ and 0.28) and POLR2A ($R^2 = 0.55$ and 0.85; Fig 5A to 5F). Interestingly, changes in H3K23ac did not correlate with changes in RNA levels in proliferating $Kat6a^{-/-}$ vs. $Kat6a^{+/+}$ NSPCs (Fig 5G and 5H).

Similarly, in differentiating $Kat6a^{-/-}$ vs. $Kat6a^{+/+}$ NSPCs, changes at promoters in H3K9ac ($R^2 = 0.61$), H3K4me3 ($R^2 = 0.56$), H3K14ac ($R^2 = 0.42$) and POLR2A ($R^2 = 0.41$) correlated positively with change in RNA levels of individual genes, while H3K23ac ($R^2 = 0.10$) correlated only weakly (S5 Fig). Few changes in MLL1 occupancy reached the FDR < 0.05 cutoff in differentiating NSPCs and so in the case of MLL1 the relationship between MLL occupancy and RNA levels is merely suggestive of a possible correlation in the promoter regions of genes (S5G Fig). In the gene bodies of differentiating $Kat6a^{-/-}$ vs. $Kat6a^{+/+}$ NSPCs, changes in H3K9ac ($R^2 = 0.62$), H3K4me3 ($R^2 = 0.56$), POLR2A ($R^2 = 0.56$) and to a lesser extent H3K14ac ($R^2 = 0.29$) correlated positively with RNA levels, while H3K23ac and MLL1 did not suggest a prominent correlation (S5 Fig).

Lastly, it is worth noting that a global gene dose effect of loss of one or two alleles of $Kat6a$ was strongly and significantly observed on the total number of mapped H3K23ac reads in differentiating NSPCs, but not on any the other chromatin marks (Fig 6A and 6B).

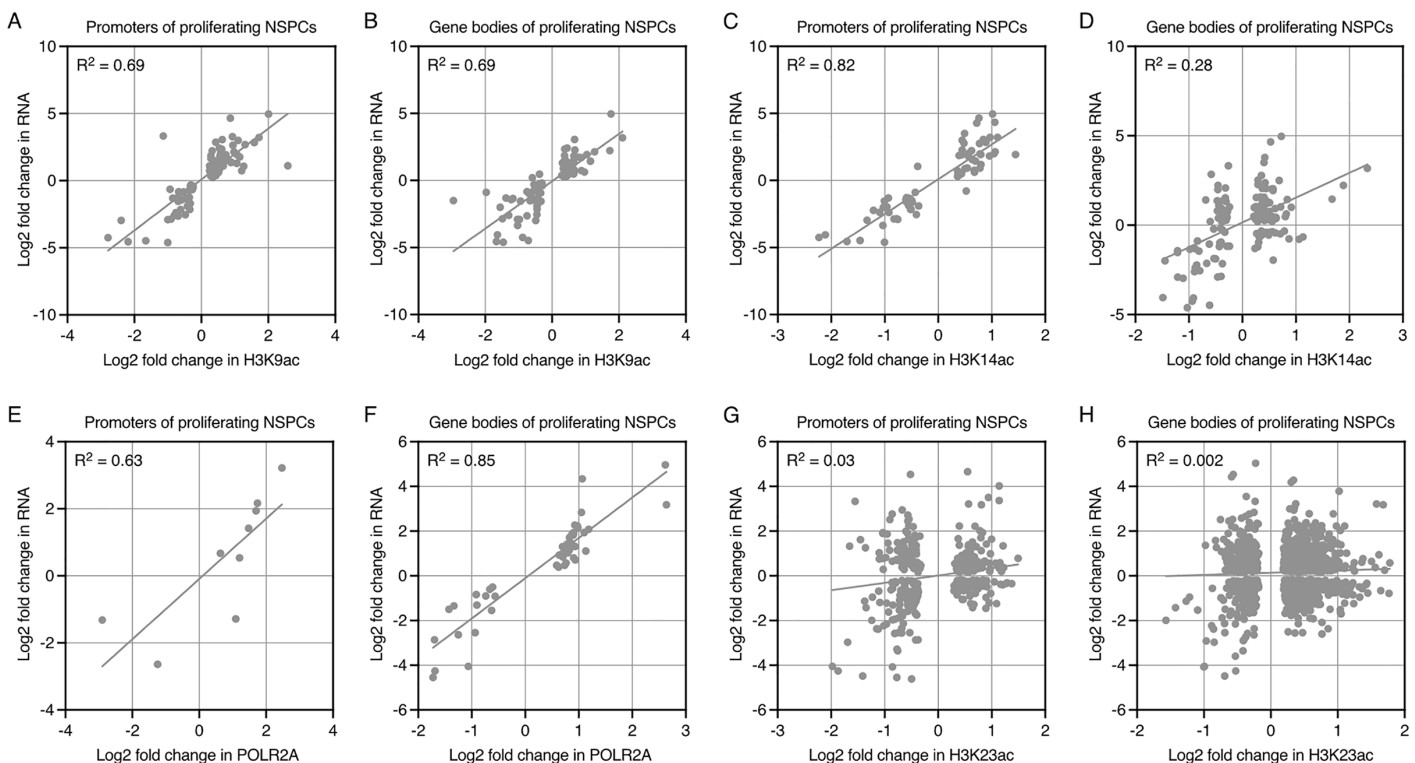

**Fig 5. Comparison of changes in histone acetylation levels and POLR2A occupancy to changes in RNA levels between proliferating $Kat6a^{+/+}$ and $Kat6a^{-/-}$ NSPCs.** (A-H) Comparison of results from RNA sequencing and CUT&Tag results of proliferating NSPCs isolated from N = 3–4 $Kat6a^{+/+}$ and 3–4 $Kat6a^{-/-}$ E12.5 embryos. Data were analysed as described in the methods section under RNA sequencing data analysis and Automated CUT&Tag sequencing data analysis. Differences in gene expression or occupancy with a false discovery rate (FDR) < 0.05 were considered significant. (A-H) Correlation between $\log_2$ fold-change in RNA levels and $\log_2$ fold-change in H3K9ac levels (A,B), H3K14ac levels (C,D), POLR2A occupancy (E,F) and H3K23ac levels (G,H) in proliferating $Kat6a^{-/-}$ vs. $Kat6a^{+/+}$ NSPCs at promoters (A,C,E,G) and bodies (B,D,F,H) of protein coding genes differentially expressed (FDR < 0.05) and differentially occupied (FDR < 0.05).

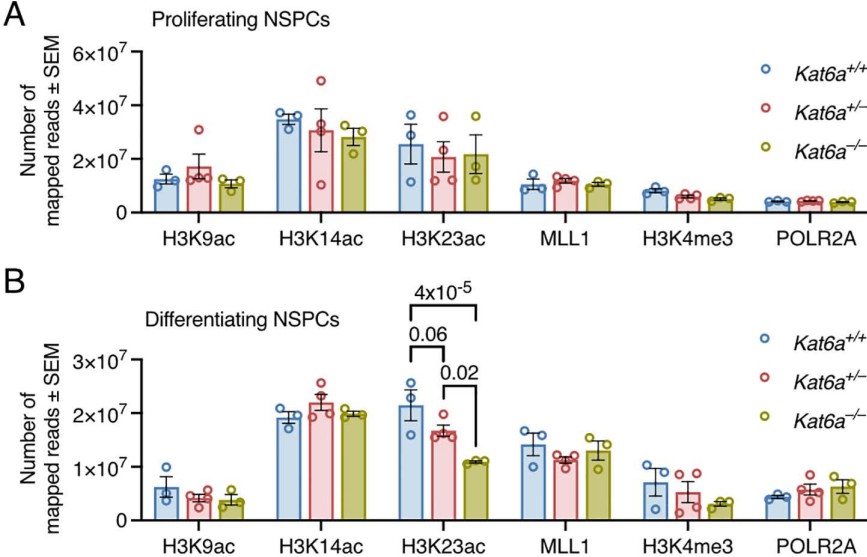

**Fig 6. The total number of mapped H3K23ac reads is reduced in differentiating *Kat6a⁻/⁻* NSPCs.** (A,B) CUT&Tag results of NSPCs isolated from N = 3 *Kat6a⁺/⁺*, 4 *Kat6a⁺/⁻* and 3 *Kat6a⁻/⁻* E12.5 embryos. Data were analysed as described in the methods section under Automated CUT&Tag sequencing data analysis. (A,B) Total number of CUT&Tag reads mapped for the histone marks and proteins indicated in proliferating (A) and differentiating (B) *Kat6a⁺/⁺*, *Kat6a⁺/⁻* and *Kat6a⁻/⁻* NSPCs. Each circle in represents NSPCs isolated from an individual mouse embryo. Data are presented as mean±SEM and were analysed by two-way ANOVA with Tukey's multiple comparison test.

## Discussion

In this study, we have shown that the absence of KAT6A causes a reduction in NSPC proliferation and neuronal differentiation. A proliferation defect has also been observed in embryonic forebrain NSC isolated from mice with a point mutation in the KAT6A histone acetyltransferase domain in a previous report by Perez-Campo and co-workers [45]. Perez-Campo and colleagues did not assess differentiation potential of the NSCs or effects on histone acetylation or gene expression. We observed changes in the expression of thousands of genes, a global reduction in H3K23ac and gene-specific reductions in H3K9ac, H3K14ac, H3K4me3, as well as reductions in occupancy by MLL1 in the absence of KAT6A. Unexpectedly, not changes in H3K23ac, but changes in H3K4me3, H3K9ac, H3K14ac, MLL1 and POLR2A were positively correlated with changes in gene expression levels when KAT6A was lacking.

The loss MYST histone acetyltransferase family members causes profound reductions in specific histone acetylation marks; loss of KAT7 (HBO1) causes the absence of H3K14ac in neural stem cells [53]; H4K16ac is eliminated in the absence of KAT8 (MOF) in mouse preimplantation embryos [54,55], and H2AZ acetylation at lysine 7 is abolished in the absences of KAT5 (TIP60) in mouse embryonic fibroblasts and human 293T and U2OS cells [56]. Congruent with our data reported here, loss of KAT6A does not result in the abolition of any histone mark, but rather a global reduction of around 50% in H3K23ac [42–44], suggesting that other acetyltransferases also mediate H3K23ac. Loss of KAT6B causes a global reduction in H3K9ac in NSPCs, E12.5 dorsal telencephalon, E16.5 cortical neurons and adult cortex and affects H3K23ac to a lesser extent [57,58].

H3K23ac is the most abundant histone acetylation mark in the mammalian genome. It occurs at 20% to 50% of all nucleosomes [59,60] and is present in genic, as well as intergenic regions [44]. Its levels within genes are positively correlated with gene expression levels [44]. In our data presented here, H3K23ac is clearly the most profoundly and the only globally affected histone acetylation target in the absence of KAT6A. While it is not reduced to zero, the reduction is highly reproducible and detectable at a genome-wide level both by western blotting and CUT&Tag. However, the reduction

in H3K23ac did not appear to correlate significantly with changes in gene expression. In contrast, the changes in MLL1 occupancy and changes in its target, H3K4me3, correlated highly and positively with changes in gene expression in the absence vs. presence of KAT6A. The absence of a correlation between changes in H3K23ac and RNA expression levels and the presence of a correlation between changes in MLL1, as well as H3K4me3, and RNA expression levels leads us to hypothesise that KAT6A has at least two functions: (1) a global function in acetylation H3K23, which we hypothesise may be required to maintain regions of the genome accessible for transcription factors and transcriptional co-regulators, and (2) facilitating the recruitment of MLL1 (and possibly other SET domain proteins) to developmental control genes such as HOX, DLX, TBX and SOX genes. Function (1) appears to be more essential in differentiating NSPCs, where we observed a *Kat6a* gene dose dependency, compared to proliferating NSPCs, possibly because another histone acetyl-transferase contributes to this role in proliferating cells. Function (2), namely affecting MLL1 occupancy at specific genes, appears to be exceptionally tightly aligned with changes in RNA levels in proliferating NSPCs. Interestingly, histone H3 poly-acetylation has been shown to enhance MLL1-mediated methylation of nucleosomal H3K4 [61], contributing to a possible mechanism. MLL1 is essential for neurogenesis and positional information in postnatal neural stem cells [62,63]. Deletion of the *Mll1* gene in the developing forebrain results in cognitive deficits and decreased expression of the home-odomain transcription factor gene *Meis2* along with decreased levels in H3K4me3 at the *Meis2* gene [64]. Congruently, we observed reduced MLL1 occupancy of the *Meis2* locus and reduced *Meis2* mRNA levels in proliferating *Kat6a*⁻/⁻ NSPCs. Suggesting a close interaction of KAT6A and MLL1, endogenous KAT6A and the endogenous C-terminal fragment of MLL1 have been reciprocally co-immunoprecipitated in HEK293T and K562 cells [65] and MLL1 occupancy of HOX genes is reduced in *Kat6a*⁻/⁻ mouse embryos [25].

Alternative to a specific effect of KAT6A on histone acetylation levels and MLL1 activity, all chromatin changes observed to correlate strongly and positively with gene expression changes could be a consequence of a yet to be discovered effect of KAT6A. However, since KAT6A has been shown to acetylate histone substrate [43,44] and MLL1 has been shown to be an order of magnitude more effective in methylating H3K4 when histone H3 is poly-acetylated [61], a mechanistic connection between KAT6A and MLL1 is likely. Remarkably, it has recently been established that the KAT6A target H3K23ac (along with H3K18ac) most robustly promotes H3K4me3 by MLL1 [66]. In this context, it is worth noting that the developmental anomalies of observed in *Kat6a*⁻/⁻ mice, namely anterior homeotic transformation, cleft palate, interrupted aortic arch and cardiac septal defects [25,39], would be curiously specific for a co-activator that acetylates H3K23 at 20% to 50% of all nucleosomes, obviously not restricted to the genes involved in these anomalies, HOX, DLX and TBX genes. In contrast, a requirement for KAT6A in recruiting MLL1 would explain this level of specificity. The role of MLL1 in regulating HOX genes is well established [67]. A role for MLL1 in DLX gene regulation has also been reported [63]. Moreover, combined mutation of *Kat6a* and the polycomb repressor complex protein encoding gene *Bmi1* rescue the homeotic transformation and HOX gene expression pattern defects observed in the individual gene mutations [23], just as combined mutation of *Mll1* and *Bmi1* do [68]. Taken together, a direct or indirect requirement for KAT6A for MLL1 recruitment and function would be congruent with data resulting from a number of studies.

KAT6A is a large multi-domain protein. Besides the centrally located MYST family-type histone lysine acetyltransferase (HAT) domain, which contains an acetyl-co-enzyme A binding pocket and a histone binding region [69], it has two amino-terminal winged helix domains, one of which has been shown to bind DNA [WHD; [70,71]], followed by two plant-homeodomain (PHD) fingers, which to bind to unmodified H3R2 and acetylated or crotonylated H3K14 [44,72,73]. Carboxyterminal of the HAT domain, KAT6A has an acidic region, a serine- and a methionine rich region. The serine- and a methionine rich region has been shown to interact with the RUNX family transcription factor, RUNX2 [74]. Given the number of protein domains, it is possible that KAT6A may directly interact with MLL proteins, which has been shown for another MYST family histone acetyltransferase, KAT8 [MOF; [75]]. Indeed, endogenous KAT6A and the endogenous C-terminal fragment of MLL1 have been reciprocally co-immunoprecipitated in HEK293T and K562 cells [65]. Alternatively, or in addition, KAT6A may interact with MLL indirectly. A physical interaction between KAT6A and the chromosomal

translocation fusion partner of MLL1, ENL, has previously been suggested in the context of leukaemia [76,77]. A strong functional cooperation between the native KAT6A and MLL1 proteins has been shown in leukaemia [78].

Loss of one allele of *Kat6a* caused no changes in gene expression that reached transcriptome-wide significance after correction for multiple testing in our NSPC datasets and in adult hippocampus [8]. Expression changes were observed after loss of one allele of *Kat6a* using single nucleus RNA sequencing of the CA3 regions of the hippocampus [8]. We observed the differential expression of 3718 and 5691 genes caused by the loss of two alleles of *Kat6a* in proliferating and differentiating NSPCs, respectively. Our observation that the *Kat6a* heterozygous expression profiles were intermediate between wild-type and homozygous profiles suggests that a multitude of subtle changes may exist in the *Kat6a* heterozygous state, which could be relevant to individuals with pathogenic variants in one allele of *KAT6A*. The effects of the complete loss of KAT6A on gene expression indicates that KAT6A is essential for the normal expression of a large number of genes, as one would expect since KAT6A is one of only 9 nuclear histone acetyltransferases [1,2], representing the capacity of the human genome to regulate approximately 20,000 protein coding genes, and non-coding genes in addition, at the level of histone acetylation. Accordingly, the defects in brain development and function that accompany Arboleda-Tham syndrome are likely to be caused by modest deregulation of many of genes, with possible contribution of multiple genes to one functional defect. For example, even just among the top 20 genes downregulated in NSPCs lacking KAT6A, four are required for neuron survival (*Esco2, Eomes, Insm1, Elavl4*), three for neuronal precursor proliferation (*Esco2, Eomes, Mycn*) and three for a normal brain size (*Eomes, Kif14, Mycn*) [79–84].

While severe brain anomalies are generally reported in mice that have lost both alleles of a specific gene and the experimental examination of the compound molecular effects of the subtle reduction in expression of thousands of genes may not be feasible, multiple subtle changes in gene expression likely underly the anomalies observed in brain function in individuals with heterozygous pathogenic variants in the *KAT6A* gene.

## Limitations of the study

The limitations of this study need to be considered and form potential topics of further investigation. First and foremost, the two-function model for KAT6A (histone acetylation and promoting directly or indirectly MLL1 function) requires further validation. The nature of the functional cooperation of KAT6A and MLL1 has not been established. This might include a direct physical interaction between KAT6A and MLL1, as reported in HEK293T and K562 cells [65], or alternatively between other members of the KAT6A and the MLL1 protein complex, which, for example, could be investigated by co-immunoprecipitation and proximity labelling experiments. Examination of the effects of loss of function of MLL1 on KAT6A recruitment and reciprocal examinations of the effects of loss of KAT6A or MLL1 on the enzymatic function of each may illuminate the relative role of recruitment and enzymatic activity and may establish causation. Another question remaining is the nature of the acetyltransferase responsible for the residual H3K23ac that is not lost in the absence of KAT6A. KAT6B contributes the H3K23ac [42,44,57,58]. However, even KAT6A and KAT6B together do not account for all H3K23ac. Triple deletion of *Kat6a*, *Kat6b* and other histone lysine acetyltransferase may arrive at a solution. A further caveat is that the RNA sequencing experiments were conducted using proliferating and differentiating NSPCs. While close to 100% of the proliferating cells expressed the stem cell marker SOX2 and no genotype effect on marker gene expression was observed, after 3 days of differentiation, significantly fewer *Kat6a*$^{-/-}$ cells expressed the neuronal marker ßIII tubulin, showing that the cell type composition was not similar between genotypes. Therefore, the RNA sequencing results of the differentiating NSPCs need to be interpreted as a combination of differences in cell type composition and direct effects of the absence of KAT6A on gene expression. This problem might be resolved using single-cell RNA sequencing in the future. Furthermore, key differentially expressed genes should be confirmed by an additional method, for example, by single-cell RNA sequencing or RT-qPCR.

The reciprocal causal relationship between histone acetylation and gene transcription has long been debated [85] but appears to be in the process of being resolved. Peak histone acetylation at some histone lysine residues temporally

precedes peak mRNA production in yeast [86] and histone acetylation facilitates de novo initiation of gene expression in mouse NSPCs [53]. Importantly, three ways of chemical inhibition of transcription has no effects on histone acetylation levels in mammalian cells [87], whereas small molecule inhibition of histone lysine acetyltransferase activity affects gene expression profoundly [44,87,88].

In conclusion, we propose a two-function model for KAT6A that includes [1] H3K23ac in large parts of the genome, which may enable access of transcription factors to a large number of genes, and [2] more specific effects on developmental control genes by facilitating or supporting the function of MLL1 directly or indirectly.

## Materials and methods

### Ethics approval

Experiments were approved by the Walter and Eliza Hall Animal Ethics Committee and were conducted in accordance with the Australian National Health and Medical Research Council Australian code for the care and use of animals for scientific purposes.

### Mice

Mice with a *Kat6a* null allele (*Kat6a⁻*), namely with the deletion of 5 exons (exons 4–8 of 17 exons in NM_001364449.1 and exons 5–9 of 18 exons in NM_001081149.2) resulting in a premature stop codon in the catalytic domain, were generated previously [25]. *Kat6a* heterozygous mice (*Kat6a⁺/⁻*) were maintained on a C57BL/6 genetic background. Mice were held in a 14-h light and 10-h dark cycle. Embryos were recovered after timed matings with noon of the day on which the vaginal plug was observed designated embryonic day 0.5 (E0.5). *Kat6a⁺/⁻* mice were intercrossed to obtain *Kat6a⁺/⁺*, *Kat6a⁺/⁻* and *Kat6a⁻/⁻* E12.5 embryos. Genotyping was performed by three-way PCR detecting a 358 bp wild type allele and 172 bp mutant allele using forward primer 5'TTC TTG ACC TCT GTG TCG TGT GC 3', wild type reverse primer 5'AGA AGT ACA GTG CTT TGG TTT CC 3', and *Kat6a* null allele reverse primer 5'ATA GGA ACT TCA TCA GTC AGG TAC 3'.

### Neural stem and progenitor cell isolation and culture

The dorsal telencephalon of E12.5 mouse embryos was dissected and placed in Eppendorf tubes on ice in 1 ml of neural stem cell (NSC) proliferation medium [Neurocult medium with the addition of proliferation supplement mouse or rat (STEMCELL Technologies, 05702), 4 µg/ml heparin solution (STEMCELL Technologies, 07980), 10 ng/ml human recombinant basic fibroblast growth factor (FGF2; STEMCELL Technologies, 78003), 20 ng/ml human recombinant epidermal growth factor (EGF; STEMCELL Technologies, 78006)]. The tissue was mechanically dissociated with a P200 pipette and filtered tip and centrifuged at 200 *g* for 5 min. The pellet was resuspended in fresh NSC proliferation medium. Cells were passed through a 70 µm cell strainer and plated onto 6 cm tissue culture Petri dishes and grown at 37°C in 5% $CO_2$ in air. Proliferating neural stem and progenitor cell (NSPC) colonies were cultured as non-adherent neurospheres, $3\times10^5$ cells were passaged every 5–6 days into 10 cm dishes. For passaging, cells were collected in 15 ml Falcon tubes and chemically dissociated with Accutase (STEMCELL Technologies, 07920), centrifuged at 200 *g* for 5 min, washed and resuspended in 2 ml of NSC proliferation medium. Live cells were quantified using trypan blue (Bio-Rad, 1450021) exclusion in an automated cell counter (Luna II, Logos Biosystems). Cells were frozen in NSC freezing medium [Neurocult medium and proliferation supplement mouse or rat, 4 µg/ml heparin solution, 10 ng/ml FGF2, 20 ng/ml EGF, 10% DMSO] at either passage 3 or passage 4 after approximately 72 h of growth without prior dissociation.

### Neural stem and progenitor cell differentiation

To induce differentiation, NSPCs were passaged after 72 h of growth, $6\times10^5$ cells were washed and resuspended in 3 ml of NSC differentiation medium [Neurocult medium and proliferation supplement mouse or rat, 1% heat inactivated embryonic

stem cell qualified foetal bovine serum (HI ES-FBS, Gibco, 30044333)] and plated onto 6 well plates precoated with 15 µg/mL poly-L-ornithine (Sigma-Aldrich, P4957) and 4.5 µg/mL laminin (Sigma-Aldrich, L2020). Differentiated cells were grown for 3 days at 37°C and 5% $CO_2$.

## Acid histone protein extraction

Histones were extracted from cultured NSPCs by acid protein extraction. Cells were collected, washed in PBS (14190144; Gibco) containing 0.5 mM sodium butyrate (B5887; Sigma-Aldrich) and cOmplete EDTA-free protease inhibitor cocktail (11873580001; Roche) and collected by centrifugation (200 $g$, 5 min). Samples were lysed in histone acid lysis buffer (10 mM HEPES pH 7.9, 1.5 mM $MgCl_2$, 10 mM KCl, and 0.5 mM DTT, supplemented with 0.5 mM sodium butyrate and cOmplete EDTA-free protease inhibitor cocktail) for 30 min at 4°C on a roller, collected by centrifugation (10,000 $g$, 10 min). The pellets were resuspended in 0.2 M $H_2SO_4$, incubated on ice for 1–2 h, centrifuged (10,000 $g$, 10 min) and the supernatant was dialyzed in dialysis tubing (Spectrum Spectra/Por Dialysis Membrane Tubing, molecular weight cut-off 20 kD; 08-607-067; Thermo Fisher Scientific) against 0.1 M acetic acid (A6283; Sigma-Aldrich) for 1 h at 4°C and MQ-$H_2O$ overnight at 4°C. Protein concentrations were determined using a bicinchoninic acid assay (23225; Thermo Fisher Scientific).

## Western immunoblotting

Acid extracted proteins were run on a 4–12% Bis–Tris gel (NP0322; Thermo Fisher Scientific) and transferred onto nitrocellulose membranes (926–31090; LI-COR Biosciences) for fluorescent detection. Membranes were blocked for 1 h at RT on a roller in blocking buffer for fluorescent detection (Intercept [PBS], 927–70001; LI-COR) and probed with antibodies against histones, namely Anti-H3K9ac (13001; Epicypher), Anti-H3K14ac (Ab53946; Abcam), Anti-H3K23ac (07–355; Millipore) and Pan-H3 (10799; Abcam) O/N at 4°C on a roller. Membranes were washed in PBS + 0.1% Tween-20 (P1379; Sigma-Aldrich) and incubated with secondary antibodies (Anti-mouse IRDye 800, 926–32210; Anti-Rabbit IRDye 680, 926–68071; both LI-COR Biosciences) for 1 h at RT on a roller. Membranes were imaged and analysed using an automated detection system (Odyssey, LI-COR Biosciences). Acetylated histone H3 marks were normalized to pan H3.

## Fluorescence activated flowcytometry

To assess the percentage of cells expressing neural stem cell, neuron, astrocyte and oligodendrocyte markers, NSPCs were grown under proliferation or differentiating conditions for 3 days. Flow cytometry was performed as described [53]. In brief, for each biological replicate, proliferating and differentiating NSPCs were performed in parallel. The proliferating NSPCs were collected in the supernatant, centrifuged at 200 $g$ for 5 min, and manually dissociated in 3 ml of culture media, then collected by centrifugation (200 $g$, 5 min) and resuspended in MT-PBS (Gibco, 10,010,023). Differentiating NSPCs underwent Accutase treatment (STEMCELL Technologies, 07920) for 3 min at 37°C. Differentiating NSPCs were quenched with differentiation medium, then collected by centrifugation (200 $g$, 5 min) and resuspended in MT-PBS. Cells were incubated at room temperature for 30 min with a LIVE/DEAD Fixable Violet Dead Cell Stain (Thermo Fisher, L34955). Cells were washed in MT-PBS and collected by centrifugation and washed in 3 ml 2% FACS buffer (2% (vol/vol) FCS, 150 mM NaCl, 3.7 mM KCl, 2.5 mM $CaCl_2$•$2H_2O$, 1.2 mM $MgSO_4$•$7H_2O$, 0.8 mM $K_2HPO_4$, 1.2 mM $KH_2PO_4$, 11.5 mM HEPES, in $H_2O$) before being collected by centrifugation. Cells were stained with conjugated antibody O4-A594 (R&D, FAB1326T) in 2% FACS buffer for 1 h on ice. Cells were washed in 3 ml FACS buffer and centrifuged (200 $g$, 5 min). Cells were fixed and permeabilised using eBioscience Intracellular fixation and permeabilization buffer set (Thermo Fisher, 88-8824-00) for 1 h on ice. Cells were washed in 3 ml 2% FACS buffer for 10 min on ice. Cells were collected by centrifugation (200 $g$, 5 min) and incubated overnight at 4°C with the following conjugated antibodies: anti-ßIII-tubulin-PerCPCy5.5 (R&D, IC1195C), anti-GFAP-A488 (Clone GA5, Invitrogen, 53-9892-82), anti-SOX2-APC (R&D, IC2018A). The following morning, cells were washed twice for 10 min in 2% FACS buffer, collected and resuspended in 2% FACS

buffer prior to analysis on a FACS analyser (LSR Fortessa X-20 Flow Cytometer; BD Biosciences) at a flow rate of less than 7,500 events/sec. Collected data were analysed using FlowJo 10.1r5 (Tree Star Inc.).

## Immunofluorescence

Differentiating NSPCs were cultured for five days in 4-well chamber slides. The culture medium was removed, cells were washed with PBS, and fixed in 4% PFA for 20 min. Cells were washed twice in PBS for 5 min each, then blocked in 10% FBS at 37°C for 1 hour. Surface marker O4 was probed against with primary antibody (Millipore, MAB345) diluted in 10% normal goat serum (NGS) for 1 hour at 37°C, followed by two washes and secondary antibody (Vector Laboratories, FI2020) under the same conditions. After two 5 min washes in PBS, primary antibodies against beta III-tubulin (Promega, G7121) and GFAP (Dako, Z0334) were co-probed in 10% NGS and 0.25% triton-X 100 for 1 hour at 37°C, washed, and re-probed with secondary antibodies (Molecular Probes, A21124 and Jackson, 111156003) diluted in 10% NGS for another hour at 37°C. The cells were washed for 5 min in PBS, after which the chamber walls were removed, and slides were mounted in Dako fluorescent mounting medium (Dako, S3023). Slides were imaged using an epifluorescence microscope (Axioplan2, Zeiss) and digital camera (AxioCam, Zeiss).

## RNA extraction and sequencing

Total RNA was extracted from NSPCs grown under proliferation or differentiating conditions for 3 days using the RNeasy mini kit (Qiagen, 74106) for proliferating cells or RNeasy micro kit (Qiagen, 74004) for differentiating cells according to the manufacturer's instructions, with the inclusion of the DNase I step. RNA concentration and quality were analysed on the Agilent 4200 Tapestation using RNA tape (Agilent, 5067–5579).

200 ng of RNA per sample was used to generate stranded sequencing libraries using the TruSeq Stranded mRNA (100 ng plus kit, Illumina) according to the manufacturer's instructions. The Samples were barcoded, pooled and sequenced on a NextSeq2000 instrument (Illumina).

## RNA sequencing analysis

After quality assessment, sequencing reads were aligned to the mm39 build of the mouse genome using Rsubread align [89]. The proportion mapped reads exceeded 99% in all samples. Gene-level counts were obtained using featureCounts with strict RefSeq mm39 annotation dated 11 April 2023 downloaded from https://bioinf.wehi.edu.au/Rsubread/annot/.

Downstream analysis was restricted to protein-coding genes only. Differentiated NSPCs and proliferating NSPCs were analysed separately. Genes with very low expression were filtered out using edgeR's filterByExpr function with default settings [90]. Sex specific genes (Xist and those unique to the Y chromosome) were removed to reduce sex biases, leaving 13,155 expressed genes for differentiated NSPCs and 12,122 expressed genes for proliferating NSPCs. Library sizes were normalised by the TMM method [91].

For the differentiated NSPCs dataset, differential expression (DE) analysis was performed using the limma-trend pipeline [92,93]. Sample-specific relative quality weights were estimated using arrayWeights [94], For the proliferating NSPCs, DE analysis was performed using the limma-voom pipeline [92,93], with correlations between samples from the same litter estimated using the duplicateCorrelation method.

Differential expression between the genotype groups was assessed using robust empirical Bayes moderated t-statistics [95]. The false discovery rate was set at 5%.

Correlations between expression profiles were displayed with limma's barcodeplot function and assessed with roast gene tests [96].

## Automated CUT&Tag sequencing

To assess multiple histone marks and genome protein occupancies in each NSPCs preparation from an individual embryo, we followed an automated CUT&Tag protocol developed by Henikoff and colleagues [47], which allows parallel

processing of many samples detecting many targets in a 96-well plate format. We used a liquid handling robot (Blue-washer) for removing liquids and adding wash buffers to plates.

For these CUT&Tag experiment, $1\times10^6$ NSPCs were cultured in a 20 cm dish for 3 days under proliferating conditions or $2\times10^6$ NSPCs were plated onto 10 cm dishes and cultured for 3 days under differentiation conditions. Both proliferating and differentiating NSPCs underwent chemical dissociation with Accutase (STEMCELL Technologies, 07920). NSPCs were collected, counted and washed. Cells were bound to concanavalin-A beads and permeabilised using 0.05% digitonin and 0.01% NP-40A. Permeabilised cells were counted, and 50,000 cells were aliquoted for each CUT&Tag reaction in a 96 well plate well. Cells were stained with primary antibody at 1:50 dilution overnight. Cells were stained with nanobody-Tn5 at 1:50 and tagmentation reactions were performed for 1 hour at 37°C. Tagmented fragments were isolated using SPRI beads. Optimal indexing PCR cycle number was determined by qPCR and all reactions for the same primary antibody were indexed using dual indexing primer [97] for 9–20 PCR cycles. Sequencing libraries concentration ranged between 0.5-15 ng/µl as determined by QuantiFluor dsDNA. Indexed sequencing libraries were pooled and sequenced on NextSeq2000 using one P3 (1.2B reads) and three P2 (400M reads) kits producing 50 bp paired end reads and 8 bp dual indexing reads and standard Illumina sequencing primer. Some samples were re-amplified using the same number of indexing PCR cycles to acquire more material for sequencing.

The antibodies to the following antigens were used: H3K9ac (Cell Signaling Technology, 9649), H3K14ac (Abcam, ab52946), H3K23ac (Millipore, 07355), MLL1 (Cell Signaling Technology, 14689), H3K4me3 (Active Motif, 36090), RNA polymerase II subunit A phosphorylated on Ser2 (POLR2A; also known as Phospho-Rpb1 CTD; Cell Signaling Technology, 13499), IgG (Epicypher, 13–0042).

## Automated CUT&Tag sequencing analysis

After sequencing-read quality assessment, reads were aligned to the mouse genome (mm39) using Rsubread 2.12.3 [89]. The proportion of uniquely mapped fragments ranged between 78% and 99% with an average of 95% in all samples. Fragment (read-pair) counts were obtained using the Rsubread's featureCounts function. Genes were defined by strict RefSeq mm39 annotation dated 08 Feb 2024 downloaded from https://bioinf.wehi.edu.au/Rsubread/annot/, and gene information was downloaded from the NCBI on 20 June 2024. Counts were obtained for gene promoters, defined as 1 kb upstream of the start of transcription (TSS) to 1 kb downstream of the TSS using Rsubread's promotorRegion function, and for gene bodies, defined as TSS to transcription end site (TES), excluding the first 1 kb downstream of the TSS to avoid an overlap with the promotor annotation. The analysis was restricted to protein-coding genes. Overlapping promoters were treated independently in the downstream analysis and hence reads mapped to overlapping promoters are assigned to each of the overlapping promoters individually.

Fragment counts were also obtained for unique H3K4me1 enrichment regions, unique H3K27ac enrichment regions, and overlapping H3K4me1 and H3K27ac enrichment regions, enriching for active enhancers, based on BED files provided as part of the GEO datasets GSM2406793 and GSM2406791. Enhancer regions overlapping with a gene or promoter region were removed and unique enhancer enrichment regions and overlapping H3K4me1 and H3K27ac enrichment regions were identified using BEDTools [98].

Genomic region counts were analysed using edgeR (v4.8.1) [90]. For each analysis, regions with low counts were filtered using filterByExpr function with default settings. Library size normalisation was performed by the cyclic loess method using edgeR's normalizeBetweenArrays.DGEList function, which implements the normalization via an offset matrix. Differential occupancy (DO) was assessed using edgeR's glmQLFit quasi-likelihood pipeline with `dispersion = NULL`, `robust = TRUE` and `abundance.trend = FALSE`. Sample quality weights were estimated using edgeR's sampleWeights function and entered as weights to glmQLFit.

## Statistical analysis

RNA sequencing data analysis methods are described in the methods section under *RNA sequencing data analysis*. CUT&Tag sequencing data analysis methods are described in the methods section under *Automated CUT&Tag*

*sequencing data analysis*. Other data were analysed using the GraphPad Prism version 7 (GraphPad Software, La Jolla, USA). The number of observations and statistical tests used are stated in the figure legends.

## Supporting information

**S1 Fig. Related to Fig 1.** Loss of KAT6A does not affect global H3K9ac or H3K14ac levels in proliferating neural stem and progenitor cells (NSPCs). (A,B) H3K9 acetylation levels and pan histone H3 levels assessed by Western blotting (A) and densitometry (B) in *Kat6a+/+* and *Kat6a−/−* NSPCs. Each lane was loaded with 0.5 µg of acid extracted protein from NSPCs isolated from an individual mouse embryo. H3K9ac levels were normalised to pan-H3 levels. (C,D) H3K14 acetylation levels and pan histone H3 levels assessed by Western blotting (C) and densitometry (D) in *Kat6a+/+* and *Kat6a−/−* NSPCs. Each lane was loaded with 2 µg of acid extracted protein from NSPCs isolated from an individual mouse embryo. H3K14ac levels were normalised to pan-H3 levels. NSPC isolates from N = 3 E12.5 embryos per genotype. Each circle in (B,D) represents NSPCs isolated from an individual mouse embryo. Data are presented as mean ± SEM and were analysed by unpaired, two-tailed Student's t-test (B,D).
(TIFF)

**S2 Fig. Related to Fig 2.** Loss of KAT6A affects gene expression in proliferating and differentiating neural stem and progenitor cells (NSPCs). (A-O) RNA sequencing data of NSPCs isolated from N = 4 *Kat6a+/+*, 4 *Kat6a+/−* and 4 *Kat6a−/−* E12.5 embryos. Data were analysed as described in the methods section under RNA sequencing data analysis. Differences in gene expression with a false discovery rate (FDR) < 0.05 were considered significant. (A,B) Multidimensional scaling plot of the leading gene expression differences between samples in pair-wise comparisons of proliferating (A) and differentiating (B) *Kat6a+/+*, *Kat6a+/−* and *Kat6a−/−* NSPC samples. (C) $Log_2$ fold-change in RNA levels of SOX genes in proliferating *Kat6a−/−* vs. *Kat6a+/+* NSPCs. FDRs shown below the bars. Genes that are downregulated or upregulated with transcriptome-wide significance are indicated with blue and red bars, respectively. Gene not changed are indicated in grey bars. (D,E) $Log_2$ fold-change in RNA levels of DLX genes (D) and MEIS genes (E) in differentiating *Kat6a−/−* and *Kat6a+/+* NSPCs. FDRs shown below the bars. (F) $Log_2$ fold-change in RNA levels of protocadherin genes in proliferating *Kat6a−/−* vs. *Kat6a+/+* NSPCs. FDRs shown above the bars. (G,H) $Log_2$ fold-change in RNA levels of the top 20 genes (by fold-change amplitude, average expression > 1 CPM) upregulated in proliferating (G) and differentiating (H) *Kat6a−/−* vs. *Kat6a+/+* NSPCs. FDRs shown above the bars. (I,K) $Log_2$ fold-change in RNA levels of genes encoding components of the KAT6A protein complex in proliferating (I) and differentiating (K) *Kat6a−/−* vs. *Kat6a+/+* NSPCs. FDRs shown below the bars. (L,M) $Log_2$ fold-change in RNA levels of histone acetyltransferase genes in proliferating (L) and differentiating (M) *Kat6a−/−* vs. *Kat6a+/+* NSPCs. FDRs shown below the bars. (N,O) Top 20 gene ontology terms (biological process) associated with genes upregulated in proliferating (N) and differentiating (O) *Kat6a−/−* vs. *Kat6a+/+* NSPCs.
(TIFF)

**S3 Fig. Related to Fig 3.** Loss of KAT6A causes broad changes in H3K9ac, H3K23ac and H3K4me3 levels as well as MLL1 occupancy that are more pronounced in differentiating than in proliferating NSPCs. (A-I) CUT&Tag results of NSPCs isolated from N = 3 *Kat6a+/+*, 4 *Kat6a+/−* and 3 *Kat6a−/−* E12.5 embryos. Data were analysed as described in the methods section under Automated CUT&Tag sequencing data analysis. Differences in occupancy with a false discovery rate (FDR) < 0.05 were considered significant. Data in (A-C,F-I) were analysed by Kruskal-Wallis test followed by Dunn's correction for multiple testing. (A-C) $Log_2$ of CUT&Tag read count per kilobase normalised to library size (RPKM) and genomic feature length, accrued over promoters (A), enhancers (B) and active enhancers (C) in differentiating *Kat6a+/+*, *Kat6a+/−* and *Kat6a−/−* NSPCs detecting H3K9ac, H3K14ac, H3K23ac, MLL1, H3K4me3 and POLR2A. (D,E) Read depth aggregates over all protein coding genes for H3K14ac (D) and POLR2A (E) over the interval from -1 kb to +1 kb of the transcription start site (TSS) in differentiating *Kat6a+/+*, *Kat6a+/−* and *Kat6a−/−* NSPCs. (F-I) $Log_2$ of CUT&Tag read count per kilobase normalised to library size (RPKM) and genomic feature length, accrued over promoters (F), enhancers (G), active enhancers (H) and gene bodies (I) in proliferating *Kat6a+/+*, *Kat6a+/−* and *Kat6a−/−* NSPCs detecting H3K9ac, H3K14ac,

H3K23ac, MLL1, H3K4me3 and POLR2A. (J-M) Read depth aggregates over all protein coding genes for H3K9ac (J), H3K23ac (K), MLL1 (L) and H3K4me3 (M) over the interval from -1 kb upstream of the transcription start site (TSS) to +1 kb downstream of the transcription end site (TES) in differentiating $Kat6a^{+/+}$, $Kat6a^{+/-}$ and $Kat6a^{-/-}$ NSPCs. Enhancers were defined as H3K4me1 enriched regions (GSM2406793) outside of promoters; active enhancers as H3K27ac (GSM2406793; [99]) and H3K4me1 (GSM2406791; [99]) enriched regions outside of promoters.
(TIFF)

**S4 Fig. Related to Fig 3.** The changes in H3K9ac, H3K23ac, H3K4me3 levels and MLL1 occupancy caused by loss of KAT6A are only modestly more pronounced in genes that are downregulated than in genes that are upregulated in differentiating and proliferating NSPCs. (A-T) CUT&Tag results of NSPCs isolated from N = 3 $Kat6a^{+/+}$, 4 $Kat6a^{+/-}$ and 3 $Kat6a^{-/-}$ E12.5 embryos. Data were analysed as described in the methods section under Automated CUT&Tag sequencing data analysis. Differences in occupancy with a false discovery rate (FDR) < 0.05 were considered significant. Data in (A-D,M-P) were analysed by Kruskal-Wallis test followed by Dunn's correction for multiple testing. Data in (Q-T) are shown as Tukey box and whisker plots and were analysed by one sample Wilcoxon test compared to a theoretical value of 1. (A-D) $Log_2$ of CUT&Tag read count per kilobase normalised to library size (RPKM) and genomic feature length, accrued over gene bodies (A,B) and promoters (C,D) of genes that are downregulated (A,C) and upregulated (D,E) based on RNA-seq results in differentiating $Kat6a^{-/-}$ vs. $Kat6a^{+/+}$NSPCs. (E-L) Read depth aggregates over protein coding genes that are downregulated (E,G,I,K) and upregulated (F,H,J,L) in differentiating $Kat6a^{+/+}$, $Kat6a^{+/-}$ and $Kat6a^{-/-}$ NSPCs for H3K9ac (E,F), H3K23ac (G,H), MLL1 (I,J) and H3K4me3 (K,L) over the interval from -1 kb to +1 kb of the transcription start site (TSS) in differentiating $Kat6a^{-/-}$ vs. $Kat6a^{+/+}$ NSPCs. (M-P) $Log_2$ of CUT&Tag read count per kilobase normalised to library size (RPKM) and genomic feature length, accrued over gene bodies (M,N) and promoters (O,P) of genes that are downregulated (M,O) and upregulated (N,P) in proliferating $Kat6a^{+/+}$, $Kat6a^{+/-}$ and $Kat6a^{-/-}$ NSPCs. (Q-T) CUT&Tag read counts ratios in genes downregulated to upregulated in $Kat6a^{-/-}$ vs. $Kat6a^{+/+}$ differentiating (Q,R) and proliferating (S,T) NSPCs (normalised to upregulated genes) in the gene bodies (Q,S) and promoters (R,T) of genes.
(TIFF)

**S5 Fig. Related to Fig 4.** Histone acetylation levels and POLR2A occupancy changes compared to changes in RNA levels between differentiating Kat6a$^{+/+}$ and Kat6a$^{-/-}$ NSPCs. (A-L) RNA sequencing and CUT&Tag sequencing results of differentiating NSPCs isolated from N = 3–4 $Kat6a^{+/+}$ and 3–4 $Kat6a^{-/-}$ E12.5 embryos. Data were analysed as described in the methods section under RNA sequencing data analysis and Automated CUT&Tag sequencing data analysis. Differences in gene expression or occupancy with a false discovery rate (FDR) < 0.05 were considered significant. (A-L) Correlation between $log_2$ fold-change in RNA levels and $log_2$ fold-change in H3K9ac (A,B), H3K14ac (C,D), H3K4me3 levels (E,F), MLL1 (G,H), POLR2A occupancy (I,J) and H3K23ac (K,L) in differentiating $Kat6a^{-/-}$ vs. $Kat6a^{+/+}$ NSPCs at promoters (A,C,E,G,I,K) and bodies (B,D,F,H,J,L) of protein coding genes differentially expressed (FDR < 0.05) and differentially occupied (FDR < 0.05), except for MLL1, which displayed too few differences at FDR < 0.05 to allow a simple linear regression, thus p < 0.001 for promoters and p < 0.01 for gene bodies were used to give an indication of the relationship of MLL1 with RNA levels in differentiating NSPCs).
(TIFF)

**S1 Raw Gels. Uncropped western blots.** This file contains the uncropped images of western blots displayed in Figs 1A, S1A and S1B.
(JPG)

**S1 Table. RNA sequencing results of proliferating $Kat6a^{-/-}$ vs. $Kat6a^{+/+}$ NSPCs.** Tab A. Differentially expressed genes in proliferating $Kat6a^{-/-}$ vs. $Kat6a^{+/+}$ NSPCs. Tab B. Gene ontology term annotation of genes differentially expressed in proliferating $Kat6a^{-/-}$ vs. $Kat6a^{+/+}$ NSPCs.
(XLSX)

**S2 Table. RNA sequencing results of differentiating *Kat6a⁻/⁻* vs. *Kat6a⁺/⁺* NSPCs.** Tab A. Differentially expressed genes in differentiating *Kat6a⁻/⁻* vs. *Kat6a⁺/⁺* NSPCs. Tab B. Gene ontology term annotation of genes differentially expressed in differentiating *Kat6a⁻/⁻* vs. *Kat6a⁺/⁺* NSPCs.
(XLSX)

**S3 Table. RNA sequencing results of proliferating *Kat6a⁺/⁻* vs. *Kat6a⁺/⁺* NSPCs.**
(XLSX)

**S4 Table. RNA sequencing results of differentiating *Kat6a⁺/⁻* vs. *Kat6a⁺/⁺* NSPCs.**
(XLSX)

**S5 Table. Auto-CUT&Tag sequencing results of proliferating *Kat6a⁻/⁻* vs. *Kat6a⁺/⁺* NSPCs Tab A.** H3K9ac – Differentially occupied genomic regions in proliferating *Kat6a⁻/⁻* vs. *Kat6a⁺/⁺* NSPCs. Tab B. H3K14ac – Differentially occupied genomic regions in proliferating *Kat6a⁻/⁻* vs. *Kat6a⁺/⁺* NSPCs. Tab C. H3K23ac – Differentially occupied genomic regions in proliferating *Kat6a⁻/⁻* vs. *Kat6a⁺/⁺* NSPCs. Tab D. MLL1 – Differentially occupied genomic regions in proliferating *Kat6a⁻/⁻* vs. *Kat6a⁺/⁺* NSPCs. Tab E. H3K4me3 – Differentially occupied genomic regions in proliferating *Kat6a⁻/⁻* vs. *Kat6a⁺/⁺* NSPCs. Tab F. POLR2A – Differentially occupied genomic regions in proliferating *Kat6a⁻/⁻* vs. *Kat6a⁺/⁺* NSPCs.
(XLSX)

**S6 Table. Auto-CUT&Tag sequencing results of differentiating *Kat6a⁻/⁻* vs. *Kat6a⁺/⁺* NSPCs.** Tab A. H3K9ac – Differentially occupied genomic regions in differentiating *Kat6a⁻/⁻* vs. *Kat6a⁺/⁺* NSPCs. Tab B. H3K14ac – Differentially occupied genomic regions in differentiating *Kat6a⁻/⁻* vs. *Kat6a⁺/⁺* NSPCs. Tab C. H3K23ac – Differentially occupied genomic regions in differentiating *Kat6a⁻/⁻* vs. *Kat6a⁺/⁺* NSPCs. Tab D. MLL1 – Differentially occupied genomic regions in differentiating *Kat6a⁻/⁻* vs. *Kat6a⁺/⁺* NSPCs. Tab E. H3K4me3 – Differentially occupied genomic regions in differentiating *Kat6a⁻/⁻* vs. *Kat6a⁺/⁺* NSPCs. Tab F. POLR2A – Differentially occupied genomic regions in differentiating *Kat6a⁻/⁻* vs. *Kat6a⁺/⁺* NSPCs.
(XLSX)

**S7 Table. Number of genes and enhancers differentially occupied in proliferating and differentiating neural stem and progenitor cells at FDR<0.05.**
(XLSX)

## Acknowledgments

The authors thank L. Johnson, L. Wilkins, S. Bound, J. Martin, R. Meeny and N. Blasch for animal care, S. Holloway and J. House for veterinary care and L. Potenza and C. Burström for excellent technical assistance.

## Author contributions

**Conceptualization:** Anne K Voss, Gordon K. Smyth, Tim Thomas.

**Data curation:** Waruni Abeysekera, Alexandra L. Garnham, Nishika Ranathunga.

**Formal analysis:** Anne K Voss, Samantha Eccles, Johannes Wichmann, Waruni Abeysekera, Alexandra L. Garnham, Nishika Ranathunga, Gordon K. Smyth.

**Funding acquisition:** Anne K Voss, Gordon K. Smyth, Tim Thomas.

**Investigation:** Samantha Eccles, Johannes Wichmann, Maria I. Bergamasco, Yuqing Yang.

**Methodology:** Johannes Wichmann.

**Supervision:** Anne K Voss, Rory Bowden, Gordon K. Smyth, Tim Thomas.

**Validation:** Samantha Eccles, Johannes Wichmann, Waruni Abeysekera, Maria I. Bergamasco, Alexandra L. Garnham, Nishika Ranathunga.

**Visualization:** Anne K Voss, Samantha Eccles, Waruni Abeysekera, Alexandra L. Garnham, Nishika Ranathunga.

**Writing – original draft:** Anne K Voss, Samantha Eccles, Johannes Wichmann, Waruni Abeysekera, Nishika Ranathunga.

**Writing – review & editing:** Anne K Voss, Gordon K. Smyth, Tim Thomas.

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
