## [Decision Letter · Decision Letter 0]

20 Feb 2026

PGENETICS-D-26-00057

KAT6A is essential for developmental control gene expression in neural stem and progenitor cells

PLOS Genetics

Dear Dr. Voss,

Thank you for submitting your manuscript to PLOS Genetics. After careful consideration, we feel that it has merit but does not fully meet PLOS Genetics's publication criteria as it currently stands. Therefore, we invite you to submit a revised version of the manuscript that addresses the points raised during the review process.

Please submit your revised manuscript within by Mar 22 2026 11:59PM. If you will need more time than this to complete your revisions, please reply to this message or contact the journal office at plosgenetics@plos.org. Please include the following items when submitting your revised manuscript:

We look forward to receiving your revised manuscript.

Kind regards,

Wei-Hsiang Huang

Academic Editor

PLOS Genetics

Paula Cohen

Section Editor

PLOS Genetics

Aimée Dudley

Editor-in-Chief

PLOS Genetics

Anne Goriely

Editor-in-Chief

PLOS Genetics

**Journal Requirements:**

At this stage, the following Authors/Authors require contributions: Anne K Voss, Samantha Eccles, Johannes Wichmann, Waruni Abeysekera, Maria Bergamasco, Alexandra Garnham, Nishika Ranathunga, Yuqing Yang, Rory Bowden, Gordon Smyth, and Tim Thomas. Please ensure that the full contributions of each author are acknowledged in the "Add/Edit/Remove Authors" section of our submission form.

The list of CRediT author contributions may be found here: https://journals.plos.org/plosgenetics/s/authorship#loc-author-contributions

2) We have noticed that you have uploaded Supporting Information files, but you have not included a list of legends. Please add a full list of legends for your Supporting Information files after the references list.

3) Please amend your detailed Financial Disclosure statement. This is published with the article. It must therefore be completed in full sentences and contain the exact wording you wish to be published.

State what role the funders took in the study. If the funders had no role in your study, please state: "The funders had no role in study design, data collection and analysis, decision to publish, or preparation of the manuscript.".

**Reviewers' comments:**

Reviewer's Responses to Questions

**Comments to the Authors:**

Reviewer #1: Voss et al. characterized the Kat6a knockout mouse with a focus on the properties of neural stem and progenitor cells (NSPCs) in in vitro differentiation settings. Homozygous loss of Kat6a resulted in slower proliferation and weaker expression of a differentiation marker in NSPCs. RNA-seq analyses revealed a set of genes affected by the loss of Kat6a. They also analyzed the epigenomic landscape of NSPCs by CUT&Tag chromatin profiling and showed that the presence of H3K9ac, H3K23ac, MLL1, and H3K4me3 was reduced by loss of Kat6a in differentiating NSPCs. By comparing the expression profile data and the epigenetic landscape data, they showed that the presence of MLL1 and H3K4me3 correlates with gene expression changes induced by Kat6a loss.

Information on the primary culture cells from the KO mice is always important and appreciated by the field. I always appreciate this group’s effort of their detailed analyses on genetically engineered animals. However, I felt the conclusions regarding what causes what are vague and the manuscript can be improved by analyzing differently. Followings are my suggestions:

Major points:

Kat6a-/- NSPCs proliferate slowly and express a subset of genes lowly and another subset of genes highly compared to the wildtype. Yet the epigenetic landscapes were analyzed as a whole in Figure 3. I would be more curious about what’s going on in the genes down-regulated by loss of Kat6a. I recommend the authors to do the analyses like Figure 3 on the gene sets of down-regulated and up-regulated separately.

I wonder how homogeneous the NSPC pools are especially in the differentiating conditions. The observed changes among the genotypes may be due to different demographics of the NSPCs. Single cell RNA-seq analysis might help.

Minor points:

On page 10, “provides an inside into major changes” may be a typo.

“insight” instead of “inside”?

In Figure 3B-E, localizations of various histone modifications and MLL1 are shown. As a comparison, the same analysis on the factors that behave similarly among the genotypes should be shown (Pol II, H3K14ac?)

Reviewer #2: Review of PGENETICS-D-26-00057:

In this manuscript, Voss et al reports the important role of mouse lysine acetyltransferase 6A (KAT6A) in regulating gene expression in neural stem and progenitor cells. For this, such cells were cultured from wild-type, heterozygous and homozygous knockout embryos at E12.5 due to lethality of homozygous knockout embryos at E13.5. Impact of Kat6a loss on histone acetylation, cell proliferation/differentiation, gene expression (via RNA-Seq) and genome-wide distribution of several histone acetylation marks (by CUT&TAG) was assessed. The results identify important roles of KAT6A in regulating the expression of genes essential for neural stem and progenitor cells, which is consistent with what is known in the literature. This study also complements nicely elegant works that have been published about KAT6A and its paralog, KAT6B, by the two groups led by Voss and Thomas, and should thus be of interests to many others working on epigenetic and development. The following issues are suggested for the authors to consider when they revise the manuscript for formal publication.

1) On page 2 (top), the statements on BRPF1, 2 and 3 are inaccurate and do not reflect the current state of knowledge, as it is typically considered by many in the field that BRPF2 and BRPF3 act through KAT7, instead KAT6A/B. The related literature should be discussed.

2) It is well established that BRPF1 serves as a key regulator of KAT6A/B, and its role in neural stem/progenitor cells has been well established. This issue can be discussed in the manuscript.

3) Because the RNA-seq data underpin many of the conclusions, independent validation (qRT–PCR of representative Sox, Dlx, Meis, or Hox targets) could be included.

4) The proposed functional interplay between KAT6A and MLL1 remains largely correlative. Experiments showing a physical or functional interaction would significantly strengthen the model and help clarify whether KAT6A acts through catalytic activity, structural roles, or both.

5) While H3K23ac is presented as a central enzymatic output, its changes do not consistently align with the transcriptional effects described.

6) The manuscript is well written, but it is always wise to double-check its text for accuracy and readability.

7) RNA-Seq and CUT&TAG data should be deposited in public databases, and the accession numbers should be included in the revised manuscript.

Reviewer #3: This study provides new insights into the role of KAT6A in neural stem and progenitor cells by integrating proliferation, differentiation, transcriptomic, and chromatin profiling data. Beyond confirming a proliferation defect, the authors demonstrate impaired neuronal differentiation and widespread transcriptional dysregulation upon Kat6a loss, accompanied by a reproducible global ~50% reduction in H3K23ac and gene-specific reductions in H3K9ac, H3K14ac, H3K4me3, MLL1 occupancy, and POLR2A. Unexpectedly, reduced KAT6A mediated H3K23ac did not correlate with expression changes, contrasted with the strong positive correlation between altered MLL1 occupancy/H3K4me3 and transcription. Therefore, the authors suggest a two-function model in which KAT6A acts both as a global H3K23 acetyltransferase maintaining chromatin accessibility and as a facilitator of MLL1 recruitment at developmental regulators such as HOX, DLX, TBX, and SOX genes (also perhaps competence for the binding of repressor proteins?). Overall, there are no major criticisms of the study as the data is of high quality. Nevertheless, the authors need to address in the discussion the limitations of their study and proposed model, and future experiments that could address these limitations. For example, how to establish a direct physical or mechanistic interaction between KAT6A and MLL1 to determine whether the observed correlation reflects causation, indirect chromatin effects, or secondary consequences of altered transcription. How to explore the basis for the global reduction, but not the complete loss of H3K23ac and the identity of compensatory acetyltransferase(s). In other words, how to functional validate the proposed two-function model. Other questions include :1) E12.5 was chosen. Would an earlier time point have been more informative i.e. to observe the primary changes in gene expression and not secondary affects given that the embryos die shortly after? 2) Regarding Figure 3A, it would be informative to show metagene plots from upstream of the TSS to down stream of the transcription termination site; 3) How do you explain the increase in Pol II in the gene body in KAT6A knockout cells (Fig. 3A); 4) For the different H3 acetylation sites, is the temporal order for their establishment known? Which acetylation sites precede transcription, and which are a consequence of transcription?

**Have all data underlying the figures and results presented in the manuscript been provided?**

Reviewer #1: Yes

Reviewer #2: Yes

Reviewer #3: Yes

PLOS authors have the option to publish the peer review history of their article (what does this mean?). If published, this will include your full peer review and any attached files.

Reviewer #1: No

Reviewer #2: No

Reviewer #3: No

**Figure resubmission:**
---

## [Decision Letter · Decision Letter 1]

20 Apr 2026

Dear Dr Voss,

We are pleased to inform you that your manuscript entitled "KAT6A is essential for developmental control gene expression in neural stem and progenitor cells" has been editorially accepted for publication in PLOS Genetics. Congratulations!

Yours sincerely,

Wei-Hsiang Huang

Academic Editor

PLOS Genetics

Paula Cohen

Section Editor

PLOS Genetics

Aimée Dudley

Editor-in-Chief

PLOS Genetics

Anne Goriely

Editor-in-Chief

PLOS Genetics

BlueSky: @plos.bsky.social

Comments from the reviewers (if applicable):

Reviewer's Responses to Questions

**Comments to the Authors:**

Reviewer #1: I have no further comments.

Many thanks to Drs Thomas and Voss for their service in science.

Reviewer #2: The authors have addressed my concerns. The paper should be of interests to many researchers and also for related patients' families.

Reviewer #3: All of my concerns were addressed.

**Have all data underlying the figures and results presented in the manuscript been provided?**

Reviewer #1: Yes

Reviewer #2: Yes

Reviewer #3: Yes

PLOS authors have the option to publish the peer review history of their article (what does this mean?). If published, this will include your full peer review and any attached files.

Reviewer #1: No

Reviewer #2: No

Reviewer #3: No

**Data Deposition**

http://datadryad.org/submit?journalID=pgenetics&manu=PGENETICS-D-26-00057R1

**Press Queries**

---

## [Editor Report · Acceptance letter]

PGENETICS-D-26-00057R1

KAT6A is essential for developmental control gene expression in neural stem and progenitor cells

Dear Dr Voss,

We are pleased to inform you that your manuscript entitled "KAT6A is essential for developmental control gene expression in neural stem and progenitor cells" has been formally accepted for publication in PLOS Genetics! Your manuscript is now with our production department and you will be notified of the publication date in due course.

With kind regards,

Anita Estes

PLOS Genetics

On behalf of:
